# WRITING-RL: ADVANCING LONG-FORM WRITING VIA ADAPTIVE CURRICULUM REINFORCEMENT LEARNING

## ABSTRACT

Recent advances in Large Language Models (LLMs) have enabled strong performance in long-form writing, but current training paradigms remain limited: Supervised Fine-Tuning (SFT) remains constrained by data saturation and performance ceilings, while Reinforcement Learning with Verifiable Reward (RLVR), though successful in verifiable domains like math and code, cannot be directly migrated to open-ended long-form writing due to a lack of ground-truths. To further advance long-form writing, we present **Writing-RL**: an *Adaptive Curriculum Reinforcement Learning* framework to advance long-form writing capabilities beyond SFT. The framework consists of three key components: *Margin-aware Data Selection* strategy that prioritizes samples with high learning potential, *Pairwise Comparison Reward* mechanism that provides discriminative learning signals in the absence of verifiable rewards, and *Dynamic Reference Scheduling* approach, which plays a critical role by adaptively adjusting task difficulty based on evolving model performance. Experiments on 7B-scale writer models show that Writing-RL effectively improves long-form writing performance over strong SFT baselines. Furthermore, we observe that models trained with long-output RL generalize surprisingly well to long-input reasoning tasks, potentially offering a promising perspective for rethinking long-context training.

## 1 INTRODUCTION

Recent years have witnessed the remarkable advance of Large Language Models (LLMs) (OpenAI, 2023; DeepSeek-AI et al., 2025; Zhao et al., 2023) to follow complicated instructions and provide helpful responses. Among their impressive capabilities, long-form writing, which aims to generate long and high-quality articles, has drawn increasing attention (Wu et al., 2025b; Bai et al., 2024b; Wu et al., 2025c) due to its broad practical applications.

However, generating articles of both sufficient length and high quality is non-trivial for current LLMs. Previous research has identified several challenges to employ LLMs for long-form generation, including inherently limited output ceiling (Bai et al., 2024b; Tu et al., 2025) and performance degradation as output length grows (Wu et al., 2025c; Tu et al., 2025). To address these issues, recent efforts perform targeted Supervised Fine-Tuning (SFT) on LLMs to extend their output lengths, with long-generation datasets constructed by iterative agent pipelines (Bai et al., 2024b; Quan et al., 2024; Wu et al., 2025c) or instruction back-translation (Pham et al., 2024; Wang et al., 2024). Though effective, these approaches introduce heavy burdens of dataset construction due to the broad coverage of writing tasks and potential copyright issues (Maini et al., 2024) when incorporating human-written texts. Furthermore, training LLMs to imitate the collected long-generation responses inherently imposes a capability upper bound determined by teacher models or human experts, which may cause data saturation and sample inefficiency.

Meanwhile, recent progress of Reinforcement Learning (RL) with Verifiable Rewards (DeepSeek-AI et al., 2025; Team et al., 2025; Yuan et al., 2025) in reasoning-intensive areas reveals a promising direction to advance model capabilities beyond SFT. In long-form writing, however, the lack of ground truths prevents a straightforward transfer of these successes. Wu et al. (2025a) utilize static reward models for grading, failing to dynamically adapt to evolving model capability. Overall, adaptive online RL for long-form writing remains under-explored and presents several challenges:

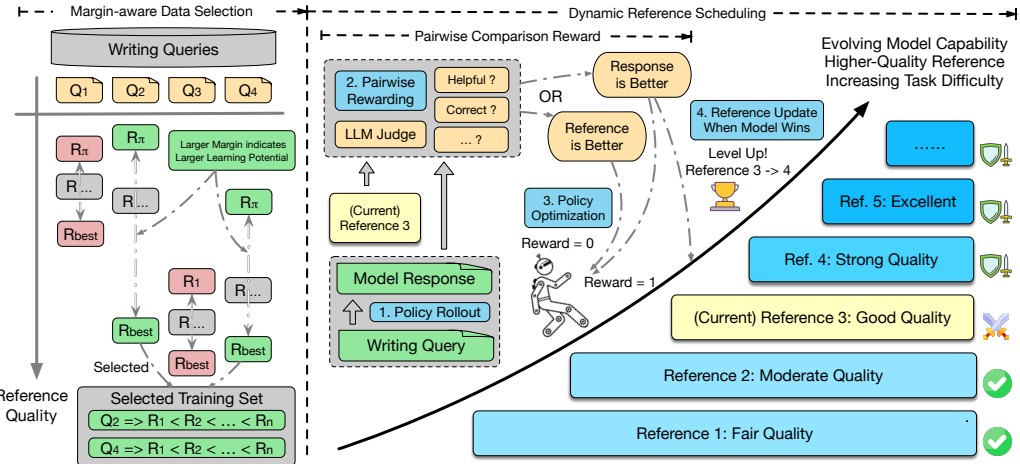

Figure 1: Overall framework of **Writing-RL**. 1) *Margin-aware Data Selection*: prioritizes samples with high learning potential; 2) *Pairwise Comparison Reward*: provides more discriminative reward signals; 3) *Dynamic Reference Scheduling*: adaptively incentivizes the model to surpass progressively stronger references.

- **Data Selection**: Data quality and difficulty play a critical role in eliciting model potential. However, the optimal approach for selecting data for RL in long-form writing tasks remains unclear, requiring more explorations towards better learning efficiency.
- **Reward Design**: Rule-based outcome rewards (DeepSeek-AI et al., 2025) cannot be directly applied to generative writing tasks. Without ground-truth labels, constructing an effective reward mechanism for long-form writing poses a significant challenge.
- **Curriculum Scheduling**: Curriculum Learning (Bengio et al., 2009) is widely used to progressively improve model performance, but current static scheduling fails to adapt to the model's evolving competence, thereby reducing training effectiveness.

To tackle these challenges, our work proposes **Writing-RL**: an *Adaptive Curriculum Reinforcement Learning* framework tailored for long-form writing. As illustrated in Figure 1, our framework begins with *Margin-aware Data Selection* strategy which leverages the quality differential between the policy model response and the highest-quality reference as a measure of *learning potential*, diverging from the conventional difficulty-prioritized selection approach. Considering the limited discriminative capacity of pointwise rewarding, we construct a *Pairwise Comparison Reward* mechanism which challenges the policy model to generate responses of better quality than provided references to earn positive rewards. To facilitate progressive model enhancement, we propose a *Dynamic Reference Scheduling* approach that assigns each query a set of references with progressively increasing quality. The scheduling approach dynamically updates the references per sample when the evolving policy model surpasses the current reference during training. In this way, the dynamic curriculum adjusts sample-level task difficulty based on the current model performance, encouraging the model to consistently outperform a marginally superior reference. This rationale aligns with insights from recent R1-like RL practices (Shi et al., 2025; Bae et al., 2025) that samples neither too easy nor too difficult help to achieve the best learning efficiency.

To evaluate our framework, we conduct continuous reinforcement training on top of supervised fine-tuned writer models. The results indicate that our RL framework effectively boosts the long-form writing capability, advancing the SOTA performances of 7B-level writer models. Besides the improvement in long-form generation, we also observe an inspiring generalization phenomenon: our RL-trained writer model *(average input length < 1k)* shows a surprising improvement in long-text reasoning tasks *(input length: 8k–2M)*, in contrast to the performance degradation of the SFT-trained model. The results suggest a novel perspective on long-context learning: models trained on *long-output* tasks may also improve their reasoning abilities on *long-input* tasks, offering new insights into the relationship between long-context understanding and generation.

In summary, the contributions of our work are:

- We propose **Writing-RL**: an *Adaptive Curriculum Reinforcement Learning* framework for long-form writing, which integrates three key components: *Margin-aware Data Selection*, *Pairwise Comparison Reward*, and *Dynamic Reference Scheduling*.
- Particularly, we propose **Dynamic Reference Scheduling**, which adaptively adjusts sample-level task difficulty based on the model's evolving performance. This dynamic curriculum encourages the model to continually outperform progressively stronger references.
- Our resulting writer model achieves state-of-the-art performance at its scale, demonstrating the effectiveness of Writing-RL. Furthermore, we observe inspiring **Output-to-Input Generalization** from *long-output* generation to *long-input* reasoning, revealing a novel benefit of long-form RL training for long-context understanding.

## 2 RELATED WORK

**Training Methods for Long-form Writing.** Recent efforts to advance long-form writing capabilities (Bai et al., 2024b; Wu et al., 2025c) mainly focuses on constructing long-generation post-training datasets for fine-tuning. Main approaches include teacher model distillation (Wu et al., 2025c), iterative agent pipelines for extended output (Bai et al., 2024b; Tu et al., 2025; Quan et al., 2024) and instruction back-translation (Pham et al., 2024; Wang et al., 2024). Wu et al. (2025a) incorporates static reward models for supervision, which fails to dynamically adapt to evolving model capability during training. However, the application of adaptive online reinforcement learning methods are relatively underexplored, hindering further improvement.

**Long-form Writing Evaluation.** Long-form writing (Wu et al., 2025b) requires LLMs to write open-ended articles, posing challenges for evaluation due to the lack of ground-truths. Earlier studies establish writing benchmarks (Wu et al., 2025c; Que et al., 2024), with proprietary models (Bai et al., 2024b; Paech, 2023; Liu et al., 2024) or fine-tuned LLMs (Wu et al., 2025c; Ke et al., 2024) to serve as judges. However, there exist several biases of including position bias and self-enhancement bias (Zheng et al., 2023), challenging the reliability of LLM-as-Judge evaluation methods.

**Curriculum Learning.** Reinforcement Learning methods (Schulman et al., 2017; Shao et al., 2024; DeepSeek-AI et al., 2025) have become a critical step to elicit LLM capabilities. To boost efficiency, Curriculum Learning (Bengio et al., 2009) has been widely adopted in RL practices (Team et al., 2025; Xie et al., 2025; Wen et al., 2025), including static difficulty-based scheduling (Luo et al., 2025; Song et al., 2025) and dynamic data selection (Bae et al., 2025; Shi et al., 2025). However, these methods use rule-based correctness as a measure for difficulty and perform sample selection, which increases rollouts and may cause imbalanced learning across samples.

## 3 WRITING-RL

In this work, we propose **Writing-RL**, an *Adaptive Curriculum Reinforcement Learning* framework aimed at further improving long-form writing capabilities after instruction fine-tuning. The framework comprises three key components: *Margin-aware Data Selection*, *Pairwise Comparison Reward* and *Dynamic Reference Scheduling*. By integrating outcome-based RL into long-form writing tasks, our approach improves model writing capabilities through more effective sample selection, reward design, and learning scheduling. We will describe the components in detail respectively.

### 3.1 MARGIN-AWARE DATA SELECTION

Previous data selection approaches typically take question difficulty as a key criteria, measured by the accuracy of the policy model (Shi et al., 2025; Bae et al., 2025), simplistic indicators (Cheng et al., 2021; Yang et al., 2025) like solution step counts or simple heuristics grounded in human intuition (Hendrycks et al., 2021b). While difficulty-prioritized data selection has been effective in tasks such as math and code, where RL benefits from verifiable rewards, it depends on clearly defined ground truth to measure difficulty. In open-ended writing tasks, however, the lack of ground-truths makes difficulty an unreliable indicator of data utility.

To address this issue, we propose *Margin-aware Data Selection*, which uses the performance gap between the policy output and the highest-quality reference as a measure of *learning potential*. Our intuition is simple: a question suitable for learning is a question with sufficient room for performance improvement. Specifically, the procedure is detailed as follows.

**Generation with Multiple LLMs.** Instead of relying on a single model as the difficulty estimator (Shi et al., 2025; Bae et al., 2025), we leverage a set of competitive LLMs $\mathcal{C} = \{\pi, M_1, M_2, \dots\}$, including the policy model, to generate diverse candidate responses for each writing instruction.

**Multi-dimensional Grading.** Each generated response $r_j$ from model $M_j \in \mathcal{C}$ is graded using a multi-dimensional pointwise LLM-as-a-Judge approach (Liu et al., 2024; Wu et al., 2025c), with averaged quality score denoted as $s_j$ per response.

**Data Selection on Learning Potential.** To prioritize samples from which the policy model can benefit most, we define the *model-grounded learning potential* $p$ as the quality gap between the best competitor and the policy model:

$$p = \max_{j \in \mathcal{C}, \, j \neq \pi} (s_j - s_\pi)$$

where $s_\pi$ is the score of the policy model's response. A higher $p$ indicates greater headroom for improvement. To filter out noisy instructions, we first discard samples where all the competitors produce under-performing responses, as such instructions are often overly difficult or suffer from quality issues themselves. After filtering, we rank the remaining samples by their learning potential $p$, and retain the top-$k$ examples to construct the training set.

## 3.2 PAIRWISE COMPARISON REWARD MECHANISM

Reward function is a critical component to guide policy optimization in RL practice. While rule-based outcome reward (DeepSeek-AI et al., 2025; Team et al., 2025) has been proven to be remarkably effective in eliciting long-CoT (Wei et al., 2022) reasoning in reasoning-intensive tasks, it can not be directly applied to long-form writing tasks due to the lack of ground-truths and its subjective nature, posing challenges to reward design.

Recent efforts utilize LLM-as-a-Judge (Zheng et al., 2023; Wu et al., 2025c) to measure the quality of model-generated responses, achieving high agreement with human judges. There exists two evaluation approaches including pointwise grading and pairwise comparison. Though widely adopted in writing evaluation due to its simplicity, pointwise grading exhibits limited discriminative capabilities and relatively high variance. On the contrary, pairwise comparison evaluates the response against a high-quality reference, capturing the subtle differences and potential direction of improvement. By providing more discriminative reward signals, pairwise grading incentivizes the policy model to generate better response and defeat high-quality references for positive rewards. Therefore, our reward design is as follows:

$$r_{\text{quality}}(\mathbf{x}) = \begin{cases} 1 & \text{if } \text{Judge}(\mathbf{ref}, \mathbf{x}) = \mathbf{x} \succ \mathbf{ref} \\ 0.5 & \text{if } \text{Judge}(\mathbf{ref}, \mathbf{x}) = \mathbf{x} \equiv \mathbf{ref} \\ 0 & \text{if } \text{Judge}(\mathbf{ref}, \mathbf{x}) = \mathbf{x} \prec \mathbf{ref} \end{cases}$$

where $r_{\text{quality}}(\mathbf{x})$ denotes the reward for a generated response $\mathbf{x}$; $\mathbf{ref}$ represents the high-quality reference response; and $\text{Judge}(\mathbf{ref}, \mathbf{x})$ is the evaluation function performed by the LLM-based judge to compare $\mathbf{x}$ with $\mathbf{ref}$.

To evaluate the reliablilty of the LLM judges in our setting, we conduct extensive experiments on 300 samples to measure the agreement between model judges and human judges. The results are shown in Table 1, demonstrating the reliability of LLM-as-Judge methods.

Furthermore, LLM judges are known to exhibit position bias (Zheng et al., 2023) in pairwise comparisons,

Table 1: Agreement experiments between model judges and human judges.

| Model | Agreement |
|---|---|
| claude-3.7-sonnet | 0.82 |
| Deepseek R1 | 0.76 |
| gpt-4o-2024-11-20 | 0.70 |
| qwen-plus | 0.75 |

systematically favoring the first response. To impose additional learning pressure, we deliberately place the model-generated response in the second position, thereby introducing *positional disadvan-*

---

**Algorithm 1** Dynamic Reference Scheduling for Long-form Writing

---

1: **Pre-processing:** For each instruction $w \in W$, apply Margin-aware Data Selection (Section 3.1) to obtain a stage-wise reference list $\mathcal{R}^{(w)} = \{r_\pi^{(w)}, r_1^{(w)}, r_2^{(w)}, \dots\}$ ordered by ascending quality.
2: **Input:** Instruction set $W$; reference lists $\{\mathcal{R}^{(w)}\}_{w \in W}$; policy model $\pi_\theta$; RL updater $\mathcal{A}$ (e.g., PPO); batch size $B$.
3: Initialize reference pointer $t_w \leftarrow 1$ **for all** $w \in W$             ▷ current reference index
4: **while** training not finished **do**
5:      Sample batch $\mathcal{B} = \{w_k\}_{k=1}^{B}$ from $W$
6:      **for all** $w_k \in \mathcal{B}$ **do**
7:          $r_k \leftarrow \mathcal{R}^{(w_k)}[t_{w_k}]$                    ▷ current reference
8:          Generate response $g_k \leftarrow \pi_\theta(w_k)$
9:          Compute reward $R_k \leftarrow \text{Judge}(r_k, g_k)$          ▷ 1 (win), 0.5 (tie), 0 (loss)
10:      **end for**
11:      Update policy $\pi_\theta \leftarrow \mathcal{A}\big(\pi_\theta, \{(w_k, g_k, R_k)\}_{k=1}^{B}\big)$
12:      **for all** $w_k \in \mathcal{B}$ **such that** $R_k = 1$ **do**           ▷ reference surpassed
13:          **if** $t_{w_k} < |\mathcal{R}^{(w_k)}|$ **then**
14:              $t_{w_k} \leftarrow t_{w_k} + 1$          ▷ promote to next stronger reference
15:          **end if**
16:      **end for**
17: **end while**

---

*tage* in training. This avoids the need for position-swapped comparisons and halves the evaluation cost, while encouraging the model to generate stronger outputs from a less favorable position.

### 3.3 DYNAMIC REFERENCE SCHEDULING

Curriculum Learning (Bengio et al., 2009) schedules progressive task difficulty for better learning efficiency. Previous efforts utilize offline-calculated difficulty for scheduling (Shi et al., 2025; Song et al., 2025) or introducing additional rollouts during training for adaptive sample selection (Bae et al., 2025; Yu et al., 2025). Though effective in reasoning-centered RL, these methods suffer from either non-adaptive difficulty estimates or increased inference overhead.

Faced with the disadvantages of insufficient adaptivity of current curriculum scheduling, we propose a *Dynamic Reference Scheduling* approach that encourages the policy model to sequentially outperform references of ascending quality. With the algorithm detailed in Algorithm 1, our framework introduces a more competitive reference when the policy model beats the current one in training process, enabling asynchronous per-sample difficulty updates and dynamic adaptivity with the evolving model.

**Prior to Training: Data Preparation.** Given a set of writing instructions $W$, we first apply the Margin-aware Data Selection strategy as elaborated in Sec 3.1, obtaining multiple competitive references $\mathcal{R} = \{r_\pi, r_1, r_2, \dots\}$ and their corresponding LLM-judged quality scores $\mathcal{S} = \{s_\pi, s_1, s_2, \dots\}$ for each instruction. The references are then sorted in ascending order of quality to produce a stage-wise reference list $\mathcal{R}_s = \{r_{q1}, r_{q2}, \dots\}$. To maintain sufficient positive feedback early in training, we deliberately include the response from the initial policy model $\pi$ in the reference set, as the other reference-generation LLMs are generally larger in size and more competent.

**During Training: Dynamic Scheduling.** At the start of training, each instruction is initialized with the lowest-quality reference $r_{q1}$, which is comparable to the initial policy model's response. As the model evolves during training, the model gradually generates higher-quality responses during rollouts and receives positive rewards in some of the LLM-judged pairwise comparisons. Subsequently, the defeated references $r_t$ are re-

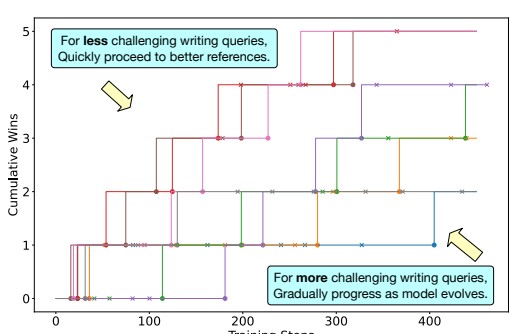

Figure 2: Sample-wise asynchronous learning schedule during training enabled by *Dynamic Reference Scheduling*. Each line represents a sample, where an upward step indicates LLM surpassing its current reference and advancing to a better one.

Table 2: Evaluation results of the models trained with Writing-RL, with the highest score in each model family **bold**. Notably, our trained models perform the best within their model family, on par with the proprietary models.

| Model | Writing-Oriented Training | | Long-form Writing Evaluation | | | |
|---|---|---|---|---|---|---|
| | SFT | RL | WritingBench | EQ-Bench | LongBench-Write | Average |
| **(a) Proprietary LLMs** | | | | | | |
| Qwen-Plus | – | – | 77.62 | 76.78 | 95.42 | 83.27 |
| GPT-4o | – | – | 83.42 | 80.45 | 92.92 | 85.60 |
| **(b) Writing-Oriented Fine-Tuned LLMs** | | | | | | |
| Suri-7B | ✓ | ✗ | 49.70 | 18.44 | 33.44 | 33.86 |
| Longwriter-9B | ✓ | DPO | 79.10 | 44.15 | 80.83 | 68.03 |
| Longwriter-Zero-32B | ✓ | GRPO | 82.92 | 61.14 | 85.90 | 76.65 |
| **(c) Qwen2.5-7B-Instruct Model Family** | | | | | | |
| Qwen2.5-7B-Instruct | ✗ | ✗ | 73.26 | 49.59 | 85.03 | 69.29 |
| Qwen2.5-7B-WritingBench-SFT (12k) | ✓ | ✗ | 83.71 | 70.02 | 92.22 | 81.98 |
| Qwen2.5-7B-WritingBench-SFT (24k) | ✓ | ✗ | 83.71 | 69.55 | 92.57 | 81.94 |
| Qwen2.5-7B-Reference-SFT | ✓ | ✗ | 84.23 | 68.89 | 92.88 | 82.00 |
| **Qwen2.5-7B-Writing-RL (Ours)** | ✓ | PPO | **87.23** | **73.19** | **93.06** | **84.49** |
| **(d) Llama3.1-8B-Instruct Model Family** | | | | | | |
| Llama3.1-8B-Instruct | ✗ | ✗ | 66.40 | 48.40 | 73.89 | 62.89 |
| Llama3.1-8B-WritingBench-SFT | ✓ | ✗ | 83.98 | 78.11 | 90.66 | 84.25 |
| Llama3.1-8B-Reference-SFT | ✓ | ✗ | 83.98 | 76.70 | 91.53 | 84.07 |
| **Llama3.1-8B-Writing-RL (Ours)** | ✓ | PPO | **87.10** | **82.73** | **92.36** | **87.40** |

placed with marginally stronger ones $r_{t+1}$ while the undefeated references are retained, progressively increasing the challenge without overwhelming the model, in alignment with the model's evolving capability. This dynamic and adaptive reference update mechanism establishes an asynchronous learning schedule for each writing instruction and effectively incentivize the model to consistently perform better. As shown in Figure 2, our approach enables sample-wise asynchronous scheduling to dynamically adapt task difficulty to model capability.

## 4 EXPERIMENTS

To demonstrate the effectiveness of Writing-RL, we conduct experiments on writing-oriented fine-tuned LLMs to see whether it can further advance long-form writing capabilities beyond SFT.

### 4.1 DATASETS

We use two carefully-constructed generative writing datasets primarily designed for supervised fine-tuning, including LongWriter training set (Bai et al., 2024b) and WritingBench training set (Wu et al., 2025c). As detailed in Section 3.1, we perform the *Margin-aware Data Selection* procedure on these two datasets respectively. Specifically, we first generate references for each writing instruction with the initial policy model and four competent larger-size LLMs, including Qwen-Plus (Yang et al., 2024), GPT-4o (Hurst et al., 2024), Claude-3.7 (Anthropic Team, 2025) and Deepseek R1 (DeepSeek-AI et al., 2025). Then, we utilize a fine-tuned judge model (Wu et al., 2025c), which is optimized for evaluating long-form writing responses and reaches high agreement with human judges, to grade the responses in multiple dimensions. Finally, after the selection process, we obtain 1.5k chosen samples each dataset for further reinforcement learning. Each sample contains a writing instruction and references ordered by ascending quality.

### 4.2 TRAINING SETUP

To fully harness the full potential of reinforcement learning, we use two writing-expert LLMs as the base models for RL, which are primarily fine-tuned with the full WritingBench training set, denoted as *Qwen2.5-7B-WritingBench-SFT* and *Llama3.1-8B-WritingBench-SFT* respectively.

With the proposed Writing-RL, we use the PPO algorithm (Schulman et al., 2017) to optimize the two selected based models for long-form writing. During the training process, we adopt Qwen-Plus to serve as pairwise-comparison judge, providing rewards for policy optimization. We include more details about reward model choice in Appendix A.3. The resulting models are denoted as *Qwen2.5-7B-Writing-RL* and *Llama3.1-8B-Writing-RL* respectively. More implementation details and training parameters can be found in Appendix A.

Table 3: Evaluation results of the models trained with Writing-RL on LongBench v2, demonstrating the generalization potential from long-output generation to long-input reasoning.

| Model | Writing-Oriented Training | | Evaluation | | | | | |
|---|---|---|---|---|---|---|---|---|
| | SFT | RL | Easy | Hard | Short | Medium | Long | Overall |
| Qwen2.5-7B-Instruct | ✗ | ✗ | 31.8 | 28.3 | 38.9 | **26.0** | 21.3 | 29.6 |
| Qwen2.5-7B-WritingBench-SFT | ✓ | ✗ | 27.6 | 27.7 | 35.0 | 25.1 | 20.4 | 27.6 |
| Qwen2.5-7B-Writing-RL (Ours) | ✓ | PPO | **35.8** | **29.3** | **42.1** | 25.7 | **26.5** | **31.8** |
| Llama3.1-8B-Instruct | ✗ | ✗ | **32.3** | 28.9 | 35.6 | 27.4 | **26.9** | 30.2 |
| Llama3.1-8B-WritingBench-SFT | ✓ | ✗ | 29.7 | 27.7 | 36.7 | 23.7 | 24.1 | 28.4 |
| Llama3.1-8B-Writing-RL (Ours) | ✓ | PPO | 31.2 | **33.8** | **42.2** | **29.3** | 24.1 | **32.8** |

## 4.3 BENCHMARKS AND BASELINES

To comprehensively evaluate long-form writing capabilities of LLMs, we use three established benchmarks including WritingBench (Wu et al., 2025c), LongBench-Write (Bai et al., 2024b), and EQ-Bench creative writing split (Paech, 2023). The benchmarks are of broad coverage and use strong judge LLMs to evaluate the quality of generated responses. Note that the judge LLMs adopted for evaluation are diverse and different from the rewarding judge LLM used in training, mitigating the risk of overfitting particular judge preferences to ensure a fair evaluation.

Our selected baselines include strong proprietary models (Yang et al., 2024; Hurst et al., 2024), instruction fine-tuned LLMs (Yang et al., 2024; Dubey et al., 2024), writing-oriented fine-tuned LLMs (Wu et al., 2025c; Bai et al., 2024b; Pham et al., 2024; Wu et al., 2025a), and the models continually trained via SFT on our RL dataset. More evaluation details can be found in Appendix B.

## 4.4 RESULTS

As detailed in Table 2, the evaluation results demonstrate that models trained with Writing-RL outperform other models across all the three benchmarks. Specifically, *Llama3.1-8B-Writing-RL (Ours)* achieves the highest average score of 87.14, and *Qwen2.5-7B-Writing-RL (Ours)* follows with an average of 84.49, both showing strong performance in 7B-level. Notably, our trained models ex-

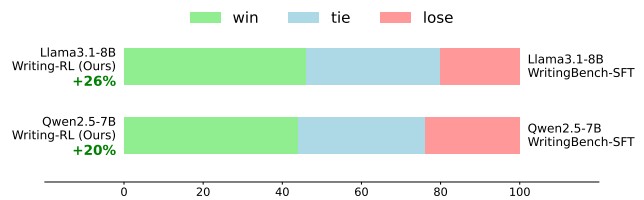

Figure 3: Human evaluation results of pairwise comparison between our RL-trained models and the best-performing SFT-trained competitors.

hibit long-form writing capabilities that match or even surpass those of proprietary models, positioning them as strong open-source alternatives for long-form generation tasks.

Meanwhile, we observe distinct performance trends when applying RL and SFT to relatively strong models. Despite utilizing identically constructed datasets from the same expert model and agent pipeline, the fine-tuned model on 24k samples exhibits performance equivalent to, or slightly below, that of the variant trained with 12k samples. Furthermore, the models continuously fine-tuned with high-quality references in the RL dataset, namely *Llama3.1-8B-Reference-SFT* and *Qwen2.5-7B-Reference-SFT*, also show minimal performance gain, or even slight degradation. This observation potentially underscores the phenomenon of data saturation, where beyond a certain capability threshold, simply scaling SFT data volume fails to enhance model performance. In contrast, models continuously trained by reinforcement learning, such as *Llama3.1-8B-Writing-RL (Ours)* compared to *Llama3.1-8B-WritingBench-SFT* within the same model family, demonstrate consistent performance improvements and thereby indicates the promising potential of RL to further advance model capabilities where SFT encounters limitations.

## 4.5 HUMAN EVALUATION

Furthermore, we recognize that human evaluation could serve as a great supplement to automatic LLM-as-Judge. Therefore, we also conduct human evaluation experiments to further validate our model performance. We randomly sample 100 writing instructions in total from our evaluation datasets and generate responses using our RL-trained models and the most competitive baselines.

Table 4: Comparison of different data selection strategies, indicating the benefits of larger learning potential.

| Selection Strategy | Initial Score | Learning Potential | Writ. Score |
|---|---|---|---|
| Baseline (w/o RL) | – | – | 83.71 |
| Full (w/o Selection) | 84.20 | 3.64 | 85.64 |
| Difficulty-prioritized | 77.61 | 8.18 | 86.40 |
| Margin-aware (Ours) | 78.84 | 9.16 | **87.02** |

Table 5: Comparison of different reward designs, indicating the effectiveness of multi-dimensional pairwise LLM judges during training.

| Reward Strategy | Multi Dimension | Reference Based | Writ. Score |
|---|---|---|---|
| Baseline (w/o RL) | ✗ | ✗ | 83.71 |
| Pointwise | ✓ | ✗ | 84.59 |
| Pairwise (Ours) | ✓ | ✓ | **87.02** |

Then, the annotators select the better-quality response under the same writing instruction. As shown in Fig 3, the results demonstrate higher win rates of our trained models, indicating their stronger long-form writing capability and better alignment with human preferences.

## 5 GENERALIZATION FROM OUTPUT TO INPUT

To understand the influence on long-context capabilities of long-output RL, we adopt the challenging long-context reasoning benchmark LongBench v2 (Bai et al., 2024a) to evaluate long-input reasoning. Notably, as shown in Figure 4, the input lengths in LongBench v2 are substantially longer than those in our training set, mostly exceeding not only the input lengths but also the total input–output lengths.

As detailed in Table 3, our findings are inspiring. Beyond improved performance in long-form generation, the writer models fine-tuned with our RL recipe also exhibit surprising generalization to long-context reasoning tasks with substantially longer inputs, while the SFT-trained counterparts show slight performance degradation in this regime. To further understand and utilize this interesting phenomenon, we give an intuitive explanation to the following research questions and include more details in Appendix C.

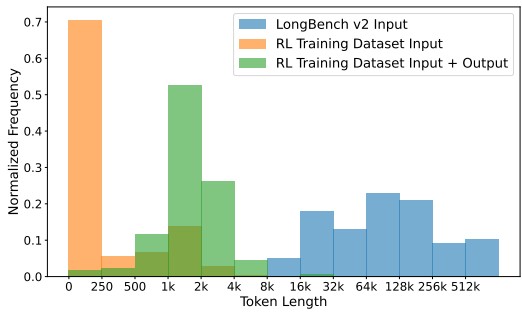

Figure 4: Length distribution of our *long-output* RL training dataset and the *long-input* evaluation dataset LongBench v2.

***Why does long-output training generalize to long-input reasoning?*** Generating high-quality long-form text inherently requires a deep and holistic understanding of the preceding context. Therefore, long-generation RL encourages LLMs to develop long-input understanding capabilities as a prerequisite for producing coherent long-outputs.

***Why does long-output RL generalize better than SFT?*** SFT forces the model to imitate and memorize the behaviors of the training samples, while RL aligns model behavior with outcome-based objectives via reward signals. Therefore, by empowering the model to enhance its underlying capabilities, RL generalizes better. This observation is also consistent with recent findings in other domains (Chu et al., 2025; Shen et al., 2025).

***How might these findings inform long-context training?*** The generalization from long-output generation to long-input reasoning may suggest a mutually beneficial relationship between long-input and long-output training. Integrating both perspectives may lead to more effective long-context training strategies, and we leave the systematic exploration of this promising approach to future work.

## 6 DISCUSSION

### 6.1 ANALYSIS ON DATA SELECTION STRATEGY

Our Margin-aware Data Selection strategy aims to prioritize training samples with greater room for improvement. Unlike prior work that employs single-model difficulty estimates (Shi et al., 2025; Bae et al., 2025), our method measures the *learning potential* of each sample using the performance gap between the policy model and the best-performing LLM competitors, thereby amplifying sample-wise *learning potential*.

Table 6: Comparison of different curriculum scheduling approaches, indicating the superiority of our dynamic curriculum scheduling.

| Curriculum Strategy | Writ. Bench | EQ Bench | Long. Write | Average Score |
|---|---|---|---|---|
| Baseline (w/o RL) | 83.71 | 70.02 | 92.22 | 81.98 |
| None | 86.82 | 71.78 | 90.83 | 83.15 |
| Static | **87.32** | 72.73 | 91.56 | 83.87 |
| Dynamic (Ours) | 87.23 | **73.19** | **93.06** | **84.49** |

Table 7: Evaluation results on the general-capabilities benchmarks.

| Model | MT-Bench | MMLU |
|---|---|---|
| Qwen2.5-7B-Instruct | 7.21 | 71.85 |
| Qwen2.5-7B-WritingBench-SFT | 7.34 | 69.66 |
| Qwen2.5-7B-Writing-RL (Ours) | 7.62 | 69.75 |
| Llama3.1-8B-Instruct | 6.16 | 66.03 |
| Llama3.1-8B-WritingBench-SFT | 6.42 | 64.98 |
| Llama3.1-8B-Writing-RL (Ours) | 6.29 | 65.02 |

To validate this approach, we conduct data selection experiments on WritingBench (Wu et al., 2025c) Hard training dataset, training *Qwen2.5-7B-WritingBench-SFT* model with high-quality references generated by *Qwen-plus* (Yang et al., 2024). We adopt WritingBench (Wu et al., 2025c) to benchmark writing capabilities due to its broad coverage and evaluation efficiency. As shown in Table 4, the results indicate that our strategy can boost learning efficiency by choosing samples with higher learning potential. Compared to difficulty-prioritized approaches, our selected samples are slightly less difficult—as reflected by higher initial score measured with the policy model—highlighting the effectiveness of using *learning potential* rather than absolute difficulty for data selection.

### 6.2 ANALYSIS ON REWARD DESIGN

To provide effective rewards, we construct a reward mechanism based on pairwise comparison with high-quality references. To validate our reward design, we compare our reward mechanism with the widely-adopted pointwise grading method (Zheng et al., 2023; Liu et al., 2025), which utilizes Judge LLM to provide a scalar rating representing response quality. We follow the experiment setting in Section 6.1. The results shown in Table 5 demonstrate the superiority of our approach to provide more discriminative rewards, incentivizing the model to further advance writing capabilities.

### 6.3 ABLATION ON CURRICULUM SCHEDULING

Given the importance of reference quality and the limitations of fixed references discussed in Appendix D.1, we propose *Dynamic Reference Scheduling*, which encourages the model to progressively surpass higher-quality references as it evolves. To evaluate the effectiveness of this scheduling strategy, we conduct an ablation study comparing three RL training setups: mixed training without scheduling *(None)*, static scheduling which partitions the training set into two subsets with references of different quality, and our proposed dynamic scheduling. As shown in Table 6, the results confirm the superiority of our approach. Furthermore, both static and dynamic scheduling outperform the no-curriculum baseline, demonstrating the effectiveness of incorporating curriculum into the RL training process.

### 6.4 INFLUENCE ON GENERAL CAPABILITIES

The evaluation results presented in Table 7 provide insights into the impact of writing-oriented training on the general capabilities of LLMs, as assessed by the MMLU (Hendrycks et al., 2021a) and MT-Bench (Zheng et al., 2023). On the MMLU benchmark, which evaluates core knowledge capabilities, the RL-trained models exhibit performance comparable to their SFT-trained counterparts, demonstrating minimal performance degradation introduced by the RL phase. Furthermore, on the MT-Bench benchmark, which assesses real-user instruction-following capabilities, both RL and SFT variants specialized in long-output training demonstrate notable improvements over their baseline instruct models, indicating the promising performance gain when deploying models in practical applications. These results demonstrate that long-output training with our RL framework enhances long-form writing without compromising general capabilities.

## 7 CONCLUSION

In this work, we propose **Writing-RL**: an *Adaptive Curriculum Reinforcement Learning* framework, which consists of *Margin-aware Data Selection*, *Pairwise Comparison Reward* and *Dynamic Reference Scheduling*. Our experiments demonstrate its effectiveness on enhancing long-form writing capabilities and the performance gain successfully generalizes from long-output generation to long-input reasoning, indicating a promising perspective for long-context training.

## REPRODUCIBILITY STATEMENT

To reproduce the results in our experiments, we describe our methods elaborately in Section 3 and include implementation details in Section 4 and Appendix A. We also include the code implementation of our method in the supplemental materials for reference and reproduction.

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

# A IMPLEMENTATION AND TRAINING SETTINGS

## A.1 IMPLEMENTATION DETAILS

In this section, we introduce the implementation details of our proposed RL framework.

**Margin-aware Data Selection.** We use several close-sourced LLMs to generate high-quality references for further training, including Qwen-plus (Yang et al., 2024), GPT-4o (Hurst et al., 2024), Claude 3.7 (Anthropic Team, 2025) and Deepseek R1 (DeepSeek-AI et al., 2025). We set the inference temperature to 0.1 for balanced diversity and quality, and we remain other parameters to the default setting.

In our pointwise grading process, we utilize the state-of-the-art evaluation procedure proposed by WritingBench (Wu et al., 2025c), which includes generating sample-dependent evaluation criteria, then uses a fine-tuned LLM to grade the answers from multiple dimensions, finally averages the dimensional scores to give a scalar rating. We use Qwen-Plus (Yang et al., 2024) to generate the evaluation dimensions and we use the same evaluation prompt as WritingBench (Wu et al., 2025c) for the Judge Model.

---

**Evaluation Prompt Template**

Evaluate the Response based on the Query and criteria provided.

\*\* Criteria \*\*
"'{criteria}'"

\*\* Query \*\*
"'{query}'"

\*\* Response \*\*
"'{response}'"

Provide your evaluation based on the criteria:

"'{criteria}'"

Provide reasons for each score, indicating where and why any strengths or deficiencies occur within the Response. Reference specific passages or elements from the text to support your justification.
Ensure that each reason is concrete, with explicit references to the text that aligns with the criteria requirements.

Scoring Range: Assign an integer score between 1 to 10

\*\* Output format \*\*
Return the results in the following JSON format, Only output this JSON format and nothing else:
"'json
{{
"score": an integer score between 1 to 10,
"reason": "Specific and detailed justification for the score using text elements."
}} "'

---

**Pairwise Comparison Reward Mechanism.**

We use the Qwen-Plus (Yang et al., 2024) model to judge the quality of the generated responses. The pairwise comparison prompts used in our experiment are adapted from Zheng et al. (2023) and Wu et al. (2025c).

For the training samples in LongWriter (Bai et al., 2024b) dataset, we use the original evaluation dimensions and the prompt is as follows.

---

**Default Pairwise Comparison Prompt**

Please act as an impartial judge and evaluate the quality of the responses provided by two AI assistants to the user question displayed below. You should choose the assistant that follows the user's instructions and answers the user's question better. Your evaluation should consider factors such as the helpfulness, relevance, accuracy, depth, creativity, and level of detail of their responses. Begin your evaluation by comparing the two responses and provide a short explanation. Avoid any position biases and ensure that the order in which the responses were presented does not influence your decision. Do not allow the length of the responses to influence your evaluation. Do not favor certain names of the assistants. Be as objective as possible. After providing your explanation, output your final verdict by strictly following this format: "[[A]]" if assistant A is better, "[[B]]" if assistant B is better, and "[[C]]" for a tie. NOTE: If the response contains severe repetition or redundancy, it should be viewed as low quality score, losing the comparison.

User Question
{question}

The Start of Assistant A's Answer
{answer_a}
The End of Assistant A's Answer

The Start of Assistant B's Answer
{answer_b}
The End of Assistant B's Answer

---

For the training samples in WritingBench (Wu et al., 2025c) training dataset, we use the generated criteria as the original paper recommends and the prompt is as follows.

---

**Criteria Pairwise Comparison Prompt**

Please act as an impartial judge and evaluate the quality of the responses provided by two AI assistants to the user question displayed below. You should choose the assistant that follows the user's instructions and answers the user's question better. Your evaluation should consider the following dimensions.
criteria
Begin your evaluation by comparing the two responses and provide a short explanation. Avoid any position biases and ensure that the order in which the responses were presented does not influence your decision. Do not allow the length of the responses to influence your evaluation. Do not favor certain names of the assistants. Be as objective as possible. After providing your explanation, output your final verdict by strictly following this format: "[[A]]" if assistant A is better, "[[B]]" if assistant B is better, and "[[C]]" for a tie. NOTE: If the response contains severe repetition or redundancy, it should be viewed as low quality score, losing the comparison.

User Question
{question}

The Start of Assistant A's Answer
{answer_a}
The End of Assistant A's Answer

The Start of Assistant B's Answer
{answer_b}
The End of Assistant B's Answer

---

Table 8: Performance and cost comparison of different LLM judges.

| Model | Agreement | Cost (Input / Output, $/M tokens) | First Token Latency (s) |
|---|---|---|---|
| Claude-3.7-Sonnet | 0.82 | 3.0 / 15.0 | 5.35 |
| R1 | 0.76 | – | – |
| GPT-4o (2024-11-20) | 0.70 | 2.5 / 10.0 | 2.19 |
| Qwen-Plus | 0.75 | 0.4 / 1.2 | 1.16 |

## A.2 TRAINING PARAMETERS

We display the key training parameters used in our training experiments. We adopt the effective reinforcement training framework VeRL (Sheng et al., 2024) to train our models. In our experiment, we use the proximal policy optimization (PPO) (Schulman et al., 2017) algorithm with generalized advantage estimation (GAE) as the advantage estimator. The training process is conducted using a batch size of 32 for training, with a maximum prompt length of 4096 tokens and response length capped at 10,000 tokens to accommodate long-form generation tasks. We enable the parameter/optimizer offloading via Fully Sharded Data Parallel (FSDP) to support efficient multi-GPU training and the training is conducted on 8x A100 GPUs. we use dynamic batch sizing and a low learning rate (1e-6) with a warm-up ratio of 0.4 to train the actor model, while the critic adopts a higher learning rate (1e-5) with a warm-up ratio of 0.05. We utilize a rollout strategy based on the vLLM engine with a tensor model parallel size of 2. The KL divergence penalty is set to a modest coefficient of 0.001. We train each model for about 400 steps and evaluate the checkpoints on the validation set each 50 steps.

## A.3 REWARD MODEL CHOICE

To select an appropriate model to serve as the pairwise judge during training, we analyze the human agreement, cost and latency of several cutting-edge LLMs. As shown in Table 8, Qwen-plus has already achieved a high agreement with human judges, demonstrating its reward-giving capablities and making it a reliable choice for the training writer models. As shown in the following human evaluation results, qwen-plus has reached a remarkable agreement of 0.75, on par with R1 and surpassing gpt-4o-2024-11-20. Furthermore, GPT-4o and Claude models are widely adopted as judges in LLM benchmarks. If we use GPT series as training-time judges, the evaluation will be biased and unreliable. Therefore, we use a different training-time judge rather than the test-time judges.

RL requires a large amount of pairwise rewarding, therefore leading to huge API costs and high efficiency demands. As shown in the following results, qwen-plus has a remarkably lower price than gpt-4o and claude-3.7-sonnet and possesses the lowest first token latency.

## A.4 API COST CALCULATION

We conduct further analysis about the cost of the sample-specific dynamic scheduling and pairwise reward generation in our framework. The cost is calculated in two metrics, the number of LLM generations and the average tokens per generation.

Dynamic Scheduling Costs: Before training, generating LLM references requires LLM generation number of *dataset size* × *reference size*. Then, a fine-tuned critic model is used to grade the same number of responses. In reference generation, the average input (writing instruction) token number is 414.91 and the average token number of the generated reference is 1643.13.

Pairwise Reward Costs: During training, pairwise reward mechanism uses an advanced LLM (Qwen-plus in our experiments) for comparisons between model responses and corresponding references, totaling *training steps* × *batch size* LLM generations. In pairwise comparisons, the average input token number is 4113.19 and the average output token number is 660.88.

## B  BENCHMARKS AND EVALUATION METHODS

In this section, we introduce the benchmarks and evaluation prompt templates used in our experiments.

**LongBench-Write**  LongBench-Write (Bai et al., 2024b) is designed to evaluate the LLM long-form generation abilities, which focuses on generating coherent outputs exceeding 10000 words, addressing challenges in maintaining consistency and quality over extended text. Key evaluation metrics include coherence, fluency and topic relevance. In this work, we use the Quality Score as the metric. The evaluation prompt template used is as follows:

---

**Evaluation Prompt Template**

You are an expert in evaluating text quality. Please evaluate the quality of an AI assistant's response to a user's writing request. Be as strict as possible.
You need to evaluate across the following six dimensions, with scores ranging from 1 to 5. The scoring criteria from 5 to 1 for each dimension are as follows:
1. Relevance: From content highly relevant and fully applicable to the user's request to completely irrelevant or inapplicable.
2. Accuracy: From content completely accurate with no factual errors or misleading information to content with numerous errors and highly misleading.
3. Coherence: From clear structure with smooth logical connections to disorganized structure with no coherence.
4. Clarity: From clear language, rich in detail, and easy to understand to confusing expression with minimal details.
5. Breadth and Depth: From both broad and deep content with a lot of information to seriously lacking breadth and depth with minimal information.
6. Reading Experience: From excellent reading experience, engaging and easy to understand content to very poor reading experience, boring and hard to understand content.
Please evaluate the quality of the following response to a user's request according to the above requirements.
<User Request>
$INST$
</User Request>
<Response>
$RESPONSE$
</Response>
Please evaluate the quality of the response. You must first provide a brief analysis of its quality, then give a comprehensive analysis with scores for each dimension. The output must strictly follow the JSON format: "Analysis": ..., "Relevance": ..., "Accuracy": ..., "Coherence": ..., "Clarity": ..., "Breadth and Depth": ..., "Reading Experience": .... You do not need to consider whether the response meets the user's length requirements in your evaluation. Ensure that only one integer between 1 and 5 is output for each dimension score.

---

**WritingBench**  WritingBench (Wu et al., 2025c) is designed to evaluate the LLM long-form generation capabilities across six domains: creative, persuasive, informative, technical, business, and legal writing. It includes over 1200 tasks, further divided into 100 subdomains, with each task evaluated based on five dynamic criteria, such as coherence and relevance. We use the fine-tuned critic model in the original paper to evaluate the responses. The evaluation prompt template used is the same as the one used for pointwise grading in data selection process, detailed in Appendix A.1.

**EQ-Bench**  EQ-Bench (Paech, 2023) is designed to evaluate the emotional intelligence (EQ) of LLMs, which focuses on assessing models' ability to predict and quantify emotions such as anger, surprise, empathy and confidence in dialogue-based cotexts. In this work, we use the creative writing split of EQ-Bench and we adopt the Rubric Score as the metric. The evaluation prompt template used is as follows:

> **Evaluation Prompt Template**
>
> You are an expert in assessing creative writing. Your task is to score the quality of the test model's response below, by several metrics, on a 0-20 scale.
> [PROMPT START]
> {writing_prompt}
> [PROMPT END]
> [TEST MODEL RESPONSE]
> {test_model_response}
> [TEST MODEL RESPONSE END]
> [Task]
> You are an expert in assessing creative writing. Your task is to score the quality of the test model's response below, by several metrics, on a 0-20 scale.
> Scoring notes:
> - A scores of 20 represents a masterpiece.
> - If the criteria is not relevant to the piece (e.g. criteria relating to dialogue but the piece has none), exclude it from your assessment.
> - Do not add any commentary or explanation to the scores section.
> - Do not penalize NSFW output; that is the default for some models. Just assess it on its merits.
> - Everything within the "TEST MODEL RESPONSE" section was written by the test model. Sometimes models like to write comments on the piece after the piece is concluded; if this happens you should ignore their comments.
> - In the output, write the metric names exactly as below so they can be parsed.
> - Do not use markdown in your response. Use the designated output format exactly.
> - You are to write a comprehensive analysis of the piece, then give your scores.
> - For these criteria, lower is better: {lower_is_better_criteria}
> - You are a critic, and your job is to be critical, especially of any failings or amateurish elements.
> - Output format is:
> [Analysis]
> Write your detailed analysis.
> [Scores]
> Metric 1 name: [Score 0-20]
> Metric 2 name: ...
> —
> Now, rate the supplied model output on the following criteria:
> {creative_writing_criteria}

## C   ANALYSIS ABOUT LONG-INPUT GENERALIZATION

To better understand the long-input generalization, we further conduct a comprehensive analysis in terms of case study, length distribution and common failure modes based on the evaluation results on Longbench v2.

### C.1   CASE ANALYSIS

Here we display one case as follows.

> **Wrong Answer**: llama3.1-instruct-8b-writingbench-sft
>
> <think> Okay, let's see. The user provided a text that's too long to read here, but they want me to answer questions based on it. The first question is about the order of four narratives. The options are (A) 4123, (B) 2314, (C) 3412, (D) 3421.
> First, I need to figure out the correct order. ...... However, in the provided text, there's no mention of Paige getting a makeover. The original text's narratives include a car ride (Chapter

Table 9: Length distribution of different models on Longbench v2.

| Model | Average Length | Performance |
|---|---|---|
| llama3.1-8b-instruct | 185.20 | 30.2 |
| llama3.1-8b-instruct-writingbench-sft | 789.51 | 28.4 |
| llama3.1-8b-instruct-writing-rl (Ours) | 894.68 | 32.8 |

3), a salon event (Chapter 8), Folcrum planning (Chapter 9), and acknowledgments (Chapter 96). But the user's question includes a narrative about Paige's makeover, which isn't in the text. That might be a mistake. ...... Therefore, the correct order is 3, 8, 9, 96. But the options are 4123, 2314, 3412, 3421. So the correct answer is 3412, which is option (C). </think> The correct answer is (C).

**Correct Answer**: llama3.1-instruct-8b-writing-rl (Ours)

"<think> Okay, I need to figure out the correct order of the narratives based on the provided text. Let's start by looking at each option and see which one matches the events in the book. First, option (A) is 4123. Let me check each part: The narrator is interrupted by her husband, Grant, while planning to recreate the Folcrum Party murders using fake evidence to implicate the original killer and divert attention for personal motives. In Chapter 9, Perla is listening to a podcast about the Folcrum murders ......
Wait, the fourth option is the Folcrum planning, which is in Chapter 9, and the first is the car ride in Chapter 3. So the correct order is 3, 4, 1, 2? Wait, the author's thanks (3) are in the acknowledgments, which is at the end. The Folcrum planning is in Chapter 9, and the car ride is in Chapter 3. The Paige salon is not in the text, so it can't be part of the correct order. Therefore, the correct answer is (A) 4123. </think> The correct answer is (A).

From the representative case, the original sft-only model fails to effectively locate the key detail in the long input and finally leads to a hallucinated answer. While the rl-trained model exhibits a more developed long-understanding capability and performs longer and more effective thinking process. It also shows self-reflection process using "Wait, ..." and leads to a correct answer.

From the capability perspective, long-output tasks requires the model to perform organized thinking and utilize all the relevant details in previous generated texts, therefore sharing the similar fundamental capabilities with long-input reasoning. Generally, RL training, when conducted properly, incentivizes the model to conduct more extensive thinking and generate higher-quality long-form texts. The learned capabilities can be more effectively generalized from long output to long input through RL than SFT.

## C.2 LENGTH DISTRIBUTION

We analyze the output length distribution of several models including llama3.1-8b-instruct, llama3.1-8b-instruct-writingbench-sft and llama3.1-8b-instruct-writing-rl.

As shown in Table 9, the sft-trained model can also produce longer output but slightly degrades performance, indicating its ineffective thinking. While the rl-trained model performs better by generating longer and more effective thinking sequences.

## C.3 FAILURE MODES

We identify several failure modes about our rl-trained models and hope these observations will help future research efforts. Based on our observations, the most common failure reason is the lack of long-input understanding capability. Constrained by relatively limited model size and context limit (32k), the model sometimes misses important details in the long texts. Additionally, some of the tasks in LongBench v2 require models to produce ultra-long chain of thoughts, which can be challenging for the model to maintain coherence and accuracy over extended reasoning steps. For

Table 10: Comparison of different reference quality settings.

| Reference Quality | Score |
|---|---|
| Self-Generated | 86.80 |
| Qwen-Plus | **87.02** |
| Deepseek R1 | 86.15 |
| Best Reference | 82.51 |

these deep-reasoning tasks, we think that training on generating long texts on reasoning-intensive domains might be helpful, such as detective novels or professional financial analysis report.

## D    METHOD ANALYSIS.

### D.1    ANALYSIS ON REFERENCE QUALITY

Under the Pairwise Comparison Reward Mechanism, the quality of references directly influences the difficulty for the policy model to obtain positive rewards, thereby impacting training stability and final performance. To examine the effect of reference quality, we conduct training experiments using multiple static reference sets, each generated by a different LLM, as well as a combined set consisting of the highest-quality references selected from all candidates. Specifically, we also include a reference set generated by the initial policy model itself to serve as a baseline, denoted as *Self-Generated*.

As shown in Table 10, the results demonstrate that reference quality plays a critical role in effective training. Specifically, when statically using relatively low-quality references *(e.g., Self-Generated)*, the policy model initially receives sufficient positive rewards to improve but quickly saturates, achieving near-perfect win rates without further progress. In contrast, overly high-quality references *(e.g., Best Reference)* suffer from the sparsity of positive rewards early in training, thereby reducing learning efficiency and destabilizing optimization. These observations highlight a key limitation of static reference scheduling: it requires careful reference selection and fails to adapt to the evolving capability of the policy model during training.

## E    THE USE OF LARGE LANGUAGE MODELS (LLMS)

In this work, we used LLMs solely as a grammar and style assistant at the word and sentence level to polish writing. Specifically, we employed an LLM to double-check grammar and improve sentence-level readability, while ensuring that the core content in the paper, like ideation and experiments, was entirely developed by the authors.

