# OpenReview forum: "Writing-RL: Advancing Long-form Writing via Adaptive Curriculum Reinforcement Learning"
_ICLR.cc/2026/Conference — ICLR 2026 Conference Withdrawn Submission_

### Official Review · Reviewer_GXgG · 2025-10-30

**Soundness:** 3
**Presentation:** 3
**Contribution:** 2
**Rating:** 4
**Confidence:** 3

**Summary:**

This work focuses on the long-form writing task. To overcome the limitations of supervised fine-tuning (SFT), the authors propose Writing-RL, an adaptive curriculum reinforcement learning framework. The framework incorporates three main components: (1) a margin-aware data selection strategy that prioritizes samples with high learning potential; (2) a comparison-based reward function that evaluates generated responses against reference texts; and (3) a dynamic reference scheduling mechanism that adjusts task difficulty over the course of training. Experimental results demonstrate the effectiveness of the framework and provide interesting insights regarding long-context input tasks.

**Strengths:**

1. Long-form text generation is an important yet relatively under-explored research problem.

2. Leveraging reinforcement learning to surpass the capacity limits of SFT is well-motivated and methodologically sound.

3. The curriculum design is intuitive and reflects principles similar to human learning behavior.

**Weaknesses:**

1. The paper claims that reference texts used during margin-aware data selection are obtained from “competitive models.” However, in the experiments, the authors employ several “competent larger models” for this role. This raises concerns about whether the observed improvements primarily stem from stronger teacher models rather than the proposed framework itself. In other words, the performance gains appear to depend heavily on the strength of the teacher model.

2. The framework relies on LLM-as-a-judge both (1) to perform pointwise evaluation for reference quality during data selection and (2) to provide pairwise comparison rewards during RL training. However, evaluating long-form text quality is inherently difficult, and LLM-based judgments are known to be unstable and sometimes inconsistent. Given that the reward signal may be noisy or unreliable, it is unclear how training stability is maintained or guaranteed.

**Questions:**

1. How sensitive is the final model performance to the choice of “competitive” (teacher) models used during margin-aware data selection? Would weaker teacher models lead to significantly worse results?

2. How do the authors ensure the reliability and consistency of multi-dimensional pointwise evaluation via LLM-as-a-judge, particularly for long-form outputs?

3. Given the known stochasticity and instability of LLM-based reward evaluations, how is reward noise mitigated to ensure stable and effective RL training?

4. Can the author explain more how to avoid the instability and randomness of using LLM-as-a-judge for generating the reward for long-form writing?

---

> ### Author Response · Authors · 2025-11-27
> **Response to Reviewer GXgG (1/n)**
>
> Thanks for your insightful review and we would like to address your concerns as follows.
>
> > [Weakness 1, Question 1] Explanations about reference generation and experiments about reference quality
>
> Thank you for the comments. We are happy to provide a more detailed explanation of our reference generation design from the following perspectives:
>
> 1. We deliberately employ strong models to generate reference responses, aim to dynamically increase the difficulty for the model to outperform these references as the model evolves. To precisely identify the source of performance gains, we conducted ablation experiments that isolate the contribution of high-quality references from that of our proposed training framework.
>
> | Model                              | WritingBench | Creative-Writing-Bench | LongBench-Write | Average |
> |------------------------------------|--------------|------------------------|-----------------|---------|
> | Qwen2.5-7B-WritingBench-SFT        | 83.71        | 70.02                  | 92.22           | 81.98   |
> | SFT with Best References           | 84.23        | 68.89                  | 92.88           | 82.00   |
> | Dynamic References SFT with RL data| 84.72        | 67.27                  | 90.83           | 80.94   |
> | Qwen2.5-7B-Writing-RL (Ours)       | 87.23        | 73.19                  | 93.06           | 84.49   |
>
> These results demonstrate that merely imitating high-quality static references yields no further improvement, indicating that SFT has already approached the performance ceiling of such imitation-based training. In contrast, the substantial gains achieved by our method stem primarily from the effectiveness of the proposed training framework.
>
> 2. To systematically assess the impact of reference quality, we conducted additional experiments using reference sets of varying quality levels.
>
> | Reference Quality             | WritingBench Score |
> |-------------------------------|--------------------|
> | Baseline (w/o RL)             | 83.71              |
> | **Initial Policy Model**         | **86.80**              |
> | Deepseek R1                   | 86.15              |
> | Highest-quality Reference     | 82.51              |
> | **Dynamic Curriculum (Ours)**     | **87.23**              |
>
> * When using weaker references (e.g., those generated by the initial policy or an earlier version of Deepseek-R1), the model still achieves meaningful gains. **Therefore, the performance gains do not heavily rely on the strong teacher models.** The limitation of relying solely on weak references is that the learning process diminishes in effectiveness during later stages, owing to the near-saturation of positive reward signals (lots of easy wins).
> * Furthermore, when using only the single highest-scoring reference per writing instruction, performance drops substantially because the reward signal becomes too sparse: the model rarely beats the reference, making effective learning extremely difficult.
> * Our full framework use an adaptive reference scheduling strategy to evolve the reference quality as the model improves, pressing the model to generate higher-quality responses throughout the training process.
>
> 3. We further compare the performance of our final trained models against both the judge model and the closed-source GPT-4o model used for generating high-quality references.
>
> | Model                              | WritingBench | Creative-Writing-Bench | LongBench-Write | Average |
> |------------------------------------|--------------|----------|------------------|---------|
> | Qwen-Plus (Judge Model)            | 77.62        | 76.78    | 95.42            | 83.27   |
> | GPT-4o (Reference Generation Model)| 83.42        | 80.45    | 92.92            | 85.60   |
> | Qwen2.5-7B-Writing-RL (Ours)       | 87.23        | 73.19      | 93.06           | 84.49   |
> | Llama3.1-8B-Writing-RL (Ours)      | 87.10        | 82.73    | 92.36            | 87.40   |
>
> The results clearly show that our best trained model successfully surpasses both the judge model (Qwen-Plus) and the reference generation model (GPT-4o). Thus, the final performance of our model is not bounded by the capabilities of either the judge or the reference-generation model. This outcome arises because, during reinforcement learning, we explicitly reward the policy for outperforming the references rather than merely imitating them.

---

> > ### Author Response · Authors · 2025-11-27
> > **Response to Reviewer GXgG (2/n)**
> >
> > > [Weakness 2, Question 2] The reliability and stability of LLM-as-Judge in **Data Selection**
> >
> > Thanks for your comments!
> >
> > For the pointwise scoring phase in data selection, we employ a fine-tuned open-source critic model(WritingBench-Critic-Model-Qwen-7B) specifically designed for evaluating writing quality as the judge. According to WritingBench[1], this model achieves 83% agreement with human judges, a level of consistency that meets or exceeds that of several proprietary closed-source model APIs, making it a reliable choice for pointwise evaluation. Moreover, when performing multi-dimensional scoring, the model assigns an independent score to one dimension per generation and then averages them to produce the final score. As the table shows, this inference-time scaling strategy has been shown to substantially improve correlation with humans. Therefore, we choose this model as a both reliable and efficient option.
> >
> > | Evaluation Metric               | Judge           | Score |
> > |----------------------------------|------------------|--------|
> > | Static Global                   | ChatGPT          | 69%   |
> > | Static Domain-Specific          | ChatGPT          | 40%   |
> > | Dynamic Query-Dependent         | ChatGPT          | 79%   |
> > | Static Global                   | Claude           | 65%   |
> > | Static Domain-Specific          | Claude           | 59%   |
> > | Dynamic Query-Dependent         | Claude           | 87%   |
> > | Dynamic Query-Dependent         | Critic Model     | **83%**   |
> >
> > To mitigate variance induced by sampling, we fix the temperature of the judge model to 0 during evaluation, thereby maximizing the deterministic stability of its outputs. Additionally, we deliberately filter out samples exhibiting extremely low or high overall scores, as well as those with excessively small standard deviations across references. This filtering step serves a dual purpose: it removes potentially problematic samples while simultaneously excluding instances that the critic model is unable to meaningfully differentiate due to its own capability limitations, thereby reducing systematic bias arising from insufficient discriminative capability on certain samples.
> >
> > | Data Selection                  | Sample Num | Policy Model Initial Score | Learning Potential | WritingBench |
> > |---------------------------------|------------|-----------------------------|--------------------|--------------|
> > | Baseline (w/o RL)               | --         | --                          | --                 | 83.71        |
> > | Full Dataset                    | 5k         | 84.20                       | 3.64               | 85.64        |
> > | Difficulty-prioritized Selection | 1.5k       | 77.61                       | 8.18               | 86.40        |
> > | Margin-aware Selection (Ours)   | 1.5k       | 78.84                       | 9.16               | 87.02        |
> >
> > Furthermore, we would like to politely remind that lots of work have confirmed the reliability and high agreement with humans when using an LLM Judge in terms of benchmark evaluation[1,2,3] and training data selection[4,5,6,7]. **As for our approach to select data based on LLM-as-Judge, we believe it is not a unique flaw of our work but is an efficient trade-off approach based on previous research given the unscalability and high expensive of human annotation.** In our practice and mannual checking, we recognize that despite the inevitable presence of some noise, our approach is effective, evidenced by our data selection ablation experiments (shown in the above table, in Section 6.1) and our final resulting model performance. We hope our experiments and analysis help to address your concerns.
> >
> > [1] Wu, Y., Mei, J., Yan, M., Li, C., Lai, S., Ren, Y., ... & Huang, F. (2025). Writingbench: A comprehensive benchmark for generative writing. arXiv.
> >
> > [2] Zheng, L., Chiang, W. L., Sheng, Y., Zhuang, S., Wu, Z., Zhuang, Y., ... & Stoica, I. (2023). Judging llm-as-a-judge with mt-bench and chatbot arena. Neurips 2023.
> >
> > [3] Liu, X., Lei, X., Wang, S., Huang, Y., Feng, A., Wen, B., ... & Tang, J. (2024, August). Alignbench: Benchmarking chinese alignment of large language models. ACL 2024.
> >
> > [4] Ye, H., & Ng, H.T. (2024). Preference-Guided Reflective Sampling for Aligning Language Models. EMNLP 2024
> >
> > [5] Fisch, A., Eisenstein, J., Zayats, V., Agarwal, A., Beirami, A., Nagpal, C., Shaw, P., & Berant, J. (2024). Robust Preference Optimization through Reward Model Distillation. Transactions on Machine Learning Research.
> >
> > [6] He, Y., Wang, H., Jiang, Z., Papangelis, A., & Zhao, H. (2024). Semi-Supervised Reward Modeling via Iterative Self-Training. EMNLP 2024
> >
> > [7] Xiao, Y., Ye, H., Chen, L., Ng, H. T., Bing, L., Li, X., & Lee, R. K. W. (2025). Finding the Sweet Spot: Preference Data Construction for Scaling Preference Optimization. arXiv.

---

> > > ### Author Response · Authors · 2025-11-27
> > > **Response to Reviewer GXgG (3/n)**
> > >
> > > > [Weakness 2, Question 3, Question 4] The reliability and stability of LLM-as-Judge in **Training Process**
> > >
> > > We are happy to provide a more detailed discussion on the **reliability** and **stability** of the reward signals used during training.
> > >
> > > **On reliability.**
> > > We conducted a rigorous human annotation study to validate the agreement between our Judge Model(Qwen-Plus) and human annotators. For each sample (consisting of an instruction and two responses, A and B), we recruited 5 independent annotators. The final human preference label was determined by majority voting among the 5 annotations. We then measured the pairwise agreement between this majority label and the judgment produced by different judge models. The results are shown below:
> > >
> > > | Model                     | Agreement with Human Majority |
> > > |---------------------------|-------------------------------|
> > > | Claude-3.5-Sonnet        | 0.82                          |
> > > | R1                        | 0.76                          |
> > > | GPT-4o (2024-11-20)       | 0.70                          |
> > > | Qwen-Plus                 | 0.75                          |
> > >
> > > These results demonstrate that **our judge model achieves relatively high consistency with human annotators**, providing reliable preference signals during training. This finding is also consistent with prior work on LLM-as-Judge [1,2,3].
> > >
> > > **On stability.**
> > > We adopted several targeted strategies to enhance the stability of the judge outputs:
> > >
> > > 1. During inference, we **fixed the temperature of the judge model to 0**, eliminating stochasticity from sampling. Previous work often employs majority voting over multiple samples, we conduct preliminary experiments (temperature = 0.3 with majority@3 or @5) on real preference pairs collected during training. The results show that majority voting tend to converge to the greedy (temperature = 0) outcome. Even when the chain-of-thought reasoning varies across samples (e.g., different ordering or weighting of evaluation dimensions), the final judgment exhibits >90% agreement with greedy decoding. Given the substantial inference cost of multiple API calls, we ultimately chose temperature = 0 as a cost-effective and sufficiently stable solution.
> > >
> > > | Majority Voting | Agreement with Greedy Decoding (%) |
> > > |-----------|--------------|
> > > | maj@1        | 87.80%       |
> > > | maj@3        | 89.43%       |
> > > | maj@5        | 91.06%       |
> > >
> > > 2. We observed the positional bias[1] in pairwise judging: models tend to favor the response in the first position. Randomizing response order would introduce additional uncertainty. To mitigate this issue, we deliberately fixed the model-generated response in the second position, imposing a **consistent position disadvantage**. This design encourages the model to produce higher-quality responses and simultaneously reduces variance caused by positional bias.
> > >
> > > 3. We explicitly provided the judge model with the **specific evaluation dimensions** relevant to each instruction, thereby reducing uncertainty arising from the the broad scope of possible judgment criteria.
> > >
> > > In summary, although LLM-as-Judge inevitably introduces some noise, the model-judge agreement with humans is high, and the model training process exhibit a certain level of robustness to moderate noise. Furthermore, *the substantial improvements in long-form writing capability and the encouraging generalization to long-input reasoning observed* in the final models provide strong empirical evidence that our judge delivered both reliable and stable reward signals throughout training.
> > >
> > > Finally, we would like to politely point out that using a reward model remains common and effective in current LLM training practices, as evidenced by ChatGPT RLHF[1], DeepSeek reward scaling[2], and Kimi K2 agentic model alignment[3], indicating that LLM-as-Judge can provide reliable and effective supervision within a certain amount of training steps when used properly. Therefore, we believe the LLM-as-Judge choice is common and therefore not a unique flaw of our proposed framework.
> > >
> > > [1] Zheng, L., ... & Stoica, I. (2023). Judging llm-as-a-judge with mt-bench and chatbot arena. Neurips 2023.
> > >
> > > [2] Liu, X., ... & Tang, J. (2024, August). Alignbench: Benchmarking chinese alignment of large language models. ACL 2024.
> > >
> > > [3] Ke, P., ... & Huang, M. (2024, August). Critiquellm: Towards an informative critique generation model for evaluation of large language model generation. ACL 2024.
> > >
> > > [4] Ouyang, L., ... & Lowe, R. (2022). Training language models to follow instructions with human feedback. Neurips 2022.
> > >
> > > [5] Liu, Z., ... & Wu, Y. (2025). Inference-time scaling for generalist reward modeling. arXiv.
> > >
> > > [6] Team, K., ... & Zhang, H. (2025). Kimi k2: Open agentic intelligence. arXiv.
> > >
> > > We agree that training stability is significant for successful RL training, and we will add these discussions in our next-version paper. Thank you for the insightful questions which helps to make this work more solid.

---

### Official Review · Reviewer_TgA2 · 2025-10-31

**Soundness:** 2
**Presentation:** 3
**Contribution:** 3
**Rating:** 4
**Confidence:** 4

**Summary:**

The paper presents a suite of techniques to improve LLM performance on a suite of writing benchmarks. The main methodological contributions are (a) selecting training examples by prioritizing examples where frontier LLMs strongly outperform the model being tuned (margin-aware selection strategy), (b) creating a pairwise reward that compares the model-generated response to the reference in deciding a score of {0, 0.5, 1}. (c) dynamically updating the references used for training examples based on the quality of model-generated outputs. Overall, as an ML problem, the authors provide sufficient evidence for the benefit of their modifications in improving benchmark performance. My main concerns are around qualitative evaluation (does the text appreciably change in style or content due to RL training, given that the improvement is only a few percentage points) and details about the human evaluation conducted - I think these are particularly important due to the nature of writing as a human-centered task.

**Strengths:**

1. Each individual method to improve performance is well motivated and intuitive, and their proposed RL training recipe leads to higher performance on both in-domain and (slightly) OOD benchmarks.

2. The analysis that RL generalizes but SFT degrades in the OOD setting is valuable and adds to the growing body of similar findings in the literature.

3. The authors perform careful ablation of many of their design choices in data selection, reward design and reference ordering (Tables 4, 5, 6), and it is expected/natural that the model faces a slight degradation in general capabilities due to the targeted fine-tuning (Table 7).

**Weaknesses:**

1. The verification of the pairwise LLM-as-judge reward is not sufficiently well documented. How many judges score each example? What is the inter-annotator agreement? Do models agree more highly on any particular label (i.e. are they less reliable on marginal 0.5 decisions)? Are models sensitive to the length of the responses being compared?

2. In Table 2, the SFT and RL models strongly outperform the baseline (unfinetuned) Qwen and Llama models in the in-domain setting. But the drop-off in generalization in Table 3 is informative. SFT is _worse_ than the base model, and RL is slightly better. Was the human eval in Section 4.5 on examples from ID or OOD benchmarks? I think we need both to get a clear picture, as well as a comparison to the base model. Also, what were the reasons behind human decisions? Were the outputs qualitatively different, or was it an improvement in faithfulness of content/coherence?

3. For a subjective task like writing, I think an open-ended case study is needed for an ecological evaluation of performance. For instance, while a benchmark score might improve by a few percentage points, when presented with suggestions from the different models, do writers actively prefer the RL-tuned model? This is different from the intrinsic human-eval in Section 4.5, which tests pairwise preference on a fixed set of examples. What I'd want to see is how much users prefer outputs from the RL-tuned model in the case where they are writing on their own and not 'judging' outputs. This could partially be mitigated by some qualitative analysis on the actual text produced - I might have missed it, but I didn't see any detailed examples of outputs from the different methods showing this.

These all seem largely fixable so I'm happy to engage during author response.

**Questions:**

1. To make the paper more self-contained, it would be helpful to include descriptions of the particular writing tasks. Also, since the tasks are created with model generation + human refinement, is it representative of the true distribution of writing tasks in terms of diversity? How is LongBench different from WriterBench? Should we expect the drop-off in performance in SFT in Table 2?

2. L.322 - based -> base, though based is unintentionally funny here

3. It would be helpful to respond to recent work that shows that LLM-as-judge is unreliable as a reward for writing [1] and that these LLM-rewards can lead to reward hacking in writing [2].

[1] Chakrabarty, Tuhin, Philippe Laban, and Chien-Sheng Wu. "Ai-slop to ai-polish? aligning language models through edit-based writing rewards and test-time computation." arXiv preprint arXiv:2504.07532 (2025).

[2] Pan, Jane, et al. "Spontaneous reward hacking in iterative self-refinement." arXiv preprint arXiv:2407.04549 (2024).

---

> ### Author Response · Authors · 2025-11-27
> **Response to Reviewer TgA2 (1/n)**
>
> Thanks for your insightful review and we would like to address your concerns as follows.
>
> > [Weakness 1] Details about verification of the pairwise LLM-as-judge reward
>
> Thanks for your constructive suggestion and we would like to present more details about the human annotation experiments.
> * We assign 5 annotators to annotate one data sample(writing instruction, response A, response B).
> * The inter-annotator agreement, with a Cohen's kappa coefficient (κ) of 0.69, is generally considered high, as values exceeding 0.6 are typically regarded as indicating relatively good agreement.
>
> In terms of the model performance on different labels, we computed pairwise inter-annotator agreement among 5 human annotators as well as the judge model’s agreement with the human-annotator majority-vote label.
>
> | Inter-annotator Agreement | Judge Model Agreement |
> |------------------|------------------|
> | <= 0.6           | 33.33%           |
> | 0.8             | 64.29%           |
> | 1.0             | 87.65%           |
>
> Results confirm the expected pattern: Difficult cases (human agreement ≤0.6) constitute only ~10% of the data and indeed show lower judge accuracy, which is natural: instances hard for humans are also harder for the model. **Across the full set, overall judge–majority agreement reaches ~80%, indicating high reliability in practice.**
>
> As for the length preference, we have conducted some analysis and the results are as follows. The results indicate that all judges (including humans) exhibit a moderate length bias. As shown below, when the two responses differ in length, the longer response wins in around 60% to 70% of cases on our annotated dataset:
>
> | Judge               | longer_win_rate (our sampled dataset) |
> |---------------------|---------------------------------------|
> | human               | 69.3                                  |
> | Claude-3.7-Sonnet   | 72.2                                  |
> | Qwen-Plus           | 67.0                                  |
> | Deepseek R1         | 66.5                                  |
> | GPT-4o-1120         | 67.2                                  |
>
> And to validate our findings, we further conduct experiments on writing samples of the LMArena public data. The results also exhibit slighter preference towards longer responses.
>
> | Judge               | longer_win_rate (sampled dataset from LMArena) |
> |---------------------|------------------------------------------------|
> | human               | 62.6                                           |
> | Claude-3.7-Sonnet  | 66.9                                           |
> | Deepseek R1         | 59.9                                           |
>
> **The slight preference of longer responses is expected and shared with human annotators**: longer responses typically contain more information and detail, making them likely to be of higher-quality in open-ended writing tasks. The model’s length preference therefore aligns well with human judgment rather than constituting an unwanted artifact.

---

> > ### Author Response · Authors · 2025-11-27
> > **Response to Reviewer TgA2 (2/n)**
> >
> > > [Weakness 2] Details about human evaluation
> >
> > Thanks for your insightful comments! The human evaluation in Section 4.5 was conducted on long-form writing tasks (ID benchmarks as you mentioned) since long-form writing tasks are open-ended and subjective, therefore requiring human efforts to assess the quality of the generated responses. And the long-input reasoning benchmark LongBench v2 contains multiple choice questions to assess long-input reasoning capabilities, therefore the results are objective and do not require humans to judge the correctness. Furthermore, we realize that conducting a case study for the generalization experiments (OOD benchmark) could be helpful to understand the phenomenon. And here we display one representative case as follows.
> >
> > Question: ... What is the order of the following events? ...
> >
> > The answer of llama3.1-instruct-8b-writingbench-sft model:
> > ```
> >     <think> Okay, let's see. The user provided a text that's too long to read here, but they want me to answer questions based on it. The first question is about the order of four narratives. First, I need to figure out the correct order. ...... However, in the provided text, there's no mention of Paige getting a makeover. The original text's narratives include a car ride (Chapter 3), a salon event (Chapter 8), Folcrum planning (Chapter 9), and acknowledgments (Chapter 96). But the user's question includes a narrative about Paige's makeover, which isn't in the text. That might be a mistake. ...... Therefore, the correct order is 3, 8, 9, 96. But the options are 4123, 2314, 3412, 3421. So the correct answer is 3412, which is option (C). </think> The correct answer is (C). [WRONG]
> > ```
> >
> > The answer of llama3.1-instruct-8b-writing-rl model (Ours):
> > ```
> >     <think> Okay, I need to figure out the correct order of the narratives based on the provided text. Let's start by looking at each option and see which one matches the events in the book. Let me check each part: The narrator is interrupted by her husband, Grant, while planning to recreate the Folcrum Party murders using fake evidence to implicate the original killer and divert attention for personal motives. In Chapter 9, Perla is listening to a podcast about the Folcrum murders ......
> >
> >     Wait, the fourth option is the Folcrum planning, which is in Chapter 9, and the first is the car ride in Chapter 3. So the correct order is 3, 4, 1, 2? Wait, the author's thanks (3) are in the acknowledgments, which is at the end. The Folcrum planning is in Chapter 9, and the car ride is in Chapter 3. The Paige salon is not in the text, so it can't be part of the correct order. Therefore, the correct answer is (A) 4123. </think> The correct answer is (A). [CORRECT]
> > ```
> >
> > From the representative case, the original sft-only model fails to effectively locate the key detail in the long input and finally leads to a hallucinated answer. While **the RL-trained model exhibits a more developed long-understanding capability and performs longer and more effective thinking process**. It also shows self-reflection process using "Wait, ..." and leads to a correct answer. From the capability perspective, long-output tasks requires the model to perform organized thinking and utilize all the relevant details in previous generated texts, therefore sharing the similar foundamental capabilities with long-input reasoning. Generally, RL training, when conducted properly, incentizes the model to conduct more extensive thinking and generate higher-quality long-form texts. The learned capabilities can be more effectively generalized from long output to long input through RL than SFT.
> >
> > As for the reasons behind the human annotators, we realize that the criteria could be potentially unclear and vary from human to human, therefore we offer clear guidance for the annotators (adapted from WritingBench). Specifically, we guide them to think like real user who poses the writing instruction and prioritize content correctness and proper use of materials after an initial annotation trial. The guidelines are as follows.
> >
> > ```
> > Please carefully read the Query, putting yourself in the user's position, and choose which of the two responses, A or B, is better. Select "A" if A is better, "B" if B is better, and "Tie" if they are about the same.
> > Do not be misled by length or formatting; focus on the **correctness of content** and the **proper use of materials** in the responses, especially if the query specifies such requirements.
> > ```
> >
> > And looking through the human evaluation results, we found that after Writing-RL, **the model excels in providing more relevant facts (better material using capability) and less factual mistakes. Furthermore, the RL-trained model responses tend to be more organized and logically coherent.**

---

> > > ### Author Response · Authors · 2025-11-27
> > > **Response to Reviewer TgA2 (3/n)**
> > >
> > > > [Weakness 3] Open case study
> > >
> > > Thanks for your insightful suggestion! We recognize that a detailed case study would be beneficial to present the performance of the Writing-RL models. And we here display a representative case study, which **uses our writer model to help a student who is preparing for the up-coming poetry recitation competition.**
> > >
> > > The writing instruction we provided to the models:
> > > ```
> > > I hope you can help me create a modern poem with the theme of "Desolation and Hope" for my Sunday recitation competition. The poem you write must incorporate the following characteristics: Use a stanza-based structure, with no more than 8 lines per stanza; The language should be concise and powerful, rich in lyricism and vivid imagery, while incorporating symbolic expressions that carry personal emotion or philosophical reflection; Centered on the theme of "Desolation and Hope," it must emphasize contrast: the opening may present symbolic scenes of desolation, the middle sections gradually introduce elements of hope, and the ending should elevate the theme with an enlightening philosophical sublimation.
> > > ```
> > >
> > > The poem generated by Writing-RL model:
> > > ```
> > > **Spring in the Cracks**
> > >
> > > I
> > > Grains of sand count tree-rings in the wind
> > > The earth splits open a thousand wounds
> > > Migratory birds’ flight paths erased by sandstorms
> > > The ancient pendulum froze in 1972
> > > Rusted iron bends into a crescent moon
> > > Every scratch an unhealed scar
> > > Only salt crystals, stubborn in the sunset
> > > crystallize into vast white fields of salt
> > >
> > > II
> > > From a fissure a tender shoot forces its way
> > > like a forgotten letter suddenly sprouting
> > > Buds tremble at the tips of dead branches
> > > measuring the sky with shivering green
> > > Children’s laughter drifts
> > > from an abandoned brickworks
> > > silver bells circling above the ruins
> > > Through gaps in corrugated roofs
> > > moss weaves green longitude and latitude
> > > with tiny root-hairs
> > > —time’s downy fur is growing
> > >
> > > III
> > > Migratory wings carry clouds
> > > and shake a few seeds loose over the wasteland
> > > Inside the concrete womb
> > > new shoots peck open petrified dreams
> > > with sharp green beaks
> > > Abandoned oil tanks become whale vertebrae
> > > vines climb the pipelines
> > > weaving silver waterfalls under moonlight
> > > Those buried seeds
> > > measure the depth of darkness
> > > with silent roots
> > > —every inch of humus
> > > is untamed freedom
> > >
> > > IV
> > > The forgotten village wakes in morning mist
> > > Willows by the old well push out fresh twigs
> > > Children’s sneakers shatter dew
> > > their laughter knocks snow from dead branches
> > > Wildflowers on the ruins
> > > turn desolation into constellations
> > > Every crack is fate’s pupil
> > > letting grains of light seep through
> > > When the first spring wind sweeps the barren plain
> > > all the dormant seeds
> > > begin writing their own rings of years
> > > —this birth inside the fissures
> > > is the earth’s deepest breath
> > > an eternal spring
> > > suddenly blooming
> > > in the corner abandoned by time
> > > ```
> > >
> > > The outputs of the original instruction-tuned model:
> > > ```
> > > **Desolation and Hope**
> > >
> > > I
> > > Withered branches sway in the wind,
> > > a boundless desolate land.
> > > Dry riverbeds, cracked earth,
> > > a deathly silence, no trace of life.
> > >
> > > II
> > > Yet right there in that barren waste,
> > > a single blade of grass tenaciously grows.
> > > It sways in the wind,
> > > stands tall beneath the scorching sun.
> > > In desolation it seeks hope,
> > > in despair it searches for life.
> > >
> > > III
> > > That desolate land
> > > was once a thriving city.
> > > Skyscrapers rose, traffic roared,
> > > now only ruins remain.
> > > But amid those ruins,
> > > people have begun to rebuild their homes.
> > > With their hands, with their sweat,
> > > with hope, with faith,
> > > they reconstruct that barren ground.
> > >
> > > IV
> > > Land of desolation, land of hope—
> > > emblem of life itself.
> > > Land of desolation, land of hope—
> > > emblem of humanity.
> > > Land of desolation, land of hope—
> > > emblem of society.
> > > Land of desolation, land of hope—
> > > emblem of the future yet to come.
> > > ```
> > >
> > > The poem generated by our Writing-RL model, *Spring in the Cracks*, surpasses the one generated by the original model through its masterful use of vivid, original imagery (salt crystallizing into white fields, green longitude and latitude, silver waterfalls”) that transforms desolation into a sensory, almost cinematic experience, while the second relies on generic, repetitive clichés (withered branches, a single blade of grass, people rebuild with sweat and hope). Ours builds a subtle, organic progression from absolute barrenness to defiant rebirth without ever stating the theme outright, trusting metaphor and surprise; the baseline spells out every moral (“this is the symbol of life/humanity/society/future”) in a didactic, slogan-like refrain that drains emotional power and kills poetic ambiguity. **From the actual user experience, our poem guides the reader’s imagination and enjoys multiple readings; the second lectures and evaporates on first contact.**

---

> > > > ### Author Response · Authors · 2025-11-27
> > > > **Response to Reviewer TgA2 (4/n)**
> > > >
> > > > > [Question 1] dataset details and SFT performance drop
> > > >
> > > > Thanks for your suggestion.
> > > > In terms of topic coverage: We randomly sampled 500 samples in our training data and each of the testsets(include all if total number is smaller than 500) and we categorized them into 8 major domains which are derived from the taxonomy of the three benchmarks. As shown below, the training distribution covers all evaluated domains broadly and the evaluation benchmarks show diverse topic distributions, contributing a more comprehensive evaluation.
> > > >
> > > > | Dataset                                | Academic & Scientific Writing | Business & Financial Writing | Legal & Policy Writing | Literary & Creative Writing | Educational & Instructional Writing | Marketing & Communications Writing | Technical & Operational Documentation | Others |
> > > > |----------------------------------------|-------------------------------|------------------------------|------------------------|-----------------------------|-------------------------------------|------------------------------------|----------------------------------|-----------------------------------|
> > > > | Training Data         | 17.00                         | 14.50                        | 13.75                  | 33.75                       | 6.75                                | 5.00                                           | 5.75                                  | 2.75 |
> > > > | WritingBench         | 11.50                         | 14.00                        | 17.50                  | 16.75                       | 19.25                               | 12.25                                   | 4.25                                  | 4.00 |
> > > > | Longbench-Write       | 17.50                         | 5.83                         | 1.67                   | 33.33                       | 15.83                               | 4.17                                          | 7.50                                  | 14.17 |
> > > > | Creative-Writing-Bench | 0.00                          | 0.00                         | 0.00                   | 100.00                      | 0.00                                | 0.00                               | 0.00                 | 0.00                                  |
> > > >
> > > > The table above demonstrates that our training distribution already spans all 8 major real-world writing domains with reasonable coverage. More importantly, the three evaluation sets deliberately target different foci (general professional writing in WritingBench, general long-output writing in LongBench-Write, and pure creative writing in Creative-Writing-Bench), achieving complementary rather than redundant coverage. This design ensures that success on three benchmarks requires genuine long-form writing ability across diverse styles and domains, far beyond memorizing any particular training distribution.
> > > >
> > > > Comparing WritingBench with LongBench-Write, WritingBench explicitly covers more professional domains that are largely absent in LongBench-Write, such as Business & Financial Writing, Legal & Policy Writing and Marketing & Communications Writing, while LongBench-Write focuses on more general creative writing scenarios.
> > > >
> > > > To further validate the influence of writing-oriented SFT on the long-input reasoning capabilities, we conduct more SFT experiments on the original SFT data, expanded SFT data with the same pipeline and SFT with our RL data.
> > > >
> > > > | Model                                     | Easy | Hard | Short | Medium | Long  | Overall |
> > > > |-------------------------------------------|------|------|-------|--------|-------|---------|
> > > > | Qwen2.5-7B-Instruct                       | 31.8 | 28.3 | 38.9  | 26.0   | 21.3  | 29.6    |
> > > > | Qwen2.5-7B-WritingBench-SFT (12k)         | 27.6 | 27.7 | 35.0  | 25.1   | 20.4  | 27.6    |
> > > > | Qwen2.5-7B-WritingBench-SFT (24k)         |  21.4	|25.4	|31.1	|21.4	|16.7  | 23.9    |
> > > > | SFT with RL data                          | 31.2 | 28.3 | 38.3  | 26.5   | 20.4  | 29.4    |
> > > > | Qwen2.5-7B-Writing-RL (Ours)              | 35.8 | 29.3 | 42.1  | 25.7   | 26.5  | 31.8    |
> > > >
> > > > The results show that continued supervised training on open-ended long-form writing data harms long-context reasoning performance. In contrast, simply exposing the model to RL data still cannot recover the full capability of the original model. Furthermore, when conducting SFT on long-form writing, continually training with data curated from the same pipeline hurts OOD performances(27.6 -> 23.9). In summary, the results indicate that **the generalization potentially stems from the learning paradigm: pure imitation learning (SFT) cannot generalize well even for the tasks both deal with long texts(long-input and long-output) and therefore share some fundamental capabilities, while RL can effectively transfer long-output capabilities towards long-input side.**
> > > >
> > > > > [Question 2] typo
> > > >
> > > > Thanks for pointing this out (with a sense of humor) and we will correct it in the next version.

---

> > > > > ### Author Response · Authors · 2025-11-27
> > > > > **Response to Reviewer TgA2 (5/n)**
> > > > >
> > > > > > [Question 3] Discussions with related work
> > > > >
> > > > > We appreciate the your suggestion to engage with recent work highlighting the unreliability of LLM-as-a-judge evaluations for writing tasks [1] and the risk of reward hacking when such judgments are used as rewards [2].
> > > > >
> > > > > ### AI-Slop to AI-Polish? Aligning Language Models through Edit-Based Writing Rewards and Test-Time Computation
> > > > >
> > > > > This work makes a valuable contribution by constructing a Writing Quality Benchmark from five subsets tailored to evaluate writing quality and assessing writing judges' reliability. On real-world writing data from LMArena, closed-source models like GPT-4o excel (highest score: 59.6), while the work's best fine-tuned judge model scores only 55.2 (below GPT-4o's 55.4). Conversely, closed-source models underperform on AI-detection datasets (e.g., Synthetic Mirror: GPT-4o: 7.5 vs. fine-tuned model: 100) and certain synthetic benchmarks. These findings indicate that **current closed-source frontier models remain reliable judges in realistic writing scenarios but falter in distinguishing AI-generated text or evaluating quality on specific synthetic data. It actually supports our approach to train a real-scenario writer model.** Additionally, *the judge prompts in the paper's Appendix case study are notably simplistic (e.g., "Which is better quality writing?")*. In our experiments, **we further improved judging reliability and human alignment by incorporating detailed guidance, sample-specific evaluation rubrics, and explicit bias warnings.**
> > > > >
> > > > > ### Spontaneous Reward Hacking in Iterative Self-Refinement
> > > > >
> > > > > The authors show that using GPT-3.5 or GPT-4 as judges for multi-round iterative writing refinement leads to spontaneous reward hacking: judge scores rise while human-evaluated quality falls. **We view this as expected but mitigable via stronger judges and refined algorithms**. The paper notes reduced hacking with GPT-4 compared to GPT-3.5, suggesting that *scaling judge capabilities helps*. Our Qwen-Plus training judge outperforms both GPT-3.5 and the early GPT-4 version used. The study also finds intensified hacking when refining the same response in shared context. To address this, *our method uses multiple reference answers as the model evolves and withholds prior generations from judges in later rounds, minimizing context-based exploitation*. We acknowledge that training under the same judge cannot continue indefinitely—hacking emerges after ~500 steps—but within controlled steps, our approach delivers substantial, consistent improvements.
> > > > >
> > > > > Finally, we would like to remind the results of our human annotation study, which show high agreement between our LLM judge and professional human annotators, supporting the validity of the reward signal used during training. These findings are corroborated by the strong performance of the resulting models.
> > > > >
> > > > > | Judge Model          | Agreement with Humans |
> > > > > |----------------------|-----------------------------|
> > > > > | Claude-3.7-Sonnet    | 0.82                        |
> > > > > | R1                   | 0.76                        |
> > > > > | GPT-4o (2024-11-20)  | 0.70                        |
> > > > > | Qwen-Plus            | 0.75                        |
> > > > >
> > > > > We sincerely thank you for recommending these two highly relevant and insightful papers. We will incorporate a detailed discussion of LLM-judge reliability and reward-hacking risks, citing both works, into the next version of our paper. This feedback has significantly helped to make our work more solid.

---

### Official Review · Reviewer_3A4d · 2025-11-01

**Soundness:** 2
**Presentation:** 2
**Contribution:** 2
**Rating:** 4
**Confidence:** 3

**Summary:**

This paper proposes an adaptive curriculum RL framework to improve long-form writing beyond SFT. The method has three components: Margin-aware data selection: select training prompts where the policy’s response lags the best competing reference by the largest margin (learning potential), estimated with multi-model generation and LLM-as-judge scoring. Pairwise comparison reward: reward the policy when its response beats a reference according to an LLM judge (win=1, tie=0.5, loss=0), with the model’s output deliberately placed in the second position to counter position bias. Dynamic reference scheduling: for each prompt, maintain an ordered list of references (increasing quality) and “level up” to a stronger reference whenever the policy beats the current one. The empirical experiment results demonstrate better performance than the baselines.

**Strengths:**

- Empirical results are good, consistent gains on three writing benchmarks across two bases, competitive with proprietary systems.


- Human evaluation corroborates automatic metrics (100 pairwise comparisons).


- Interesting generalization from long-output RL to long-input reasoning on LongBench v2.

**Weaknesses:**

- Training rewards hinge on one judge (Qwen-Plus). Although different judges are used at eval time, the training-time reward could encode judge-specific preferences. Agreement with humans is reported but limited in scope.

- Only ~1.5k prompts per source are used for RL after filtering; more detail on topic coverage and potential overlap with evaluation prompts would help assess overfitting risks.

- Tables lack variance or significance tests; it is unclear whether deltas (often 1–2 points) are statistically reliable across seeds.

- Compute and cost. The method involves multi-model reference generation and LLM judges during training; however, the training-time cost/latency is not quantified, making reproducibility/industrial adoption harder to gauge. (Setup described, but cost analysis missing.)

**Questions:**

- How does performance change if you (a) ensemble multiple judges for rewards, (b) randomize judges per batch, or (c) periodically swap the judges? Does this reduce bias and further improve generalization?

- What are the reference generation and reward-computation costs (GPU hours, $) for your typical run? A breakdown would clarify practicality.

- Ablate curriculum vs RL: If you keep SFT optimization but use dynamic reference scheduling as hard-negative distillation (no RL), how much of the gain remains? Conversely, PPO with pointwise rewards but no scheduling?

- Generalization controls: Can you report: (i) length-matched SFT (same output lengths as RL rollouts), (ii) pairwise-reward RL without margin-aware selection, and (iii) dynamic scheduling with static pointwise rewards?
- What does the color represent in Figure 2?

- In line 87, the policy model response and the highest-quality reference, how to define the high-quality references? If we use some low-quality references from some models, how about those results?

---

> ### Author Response · Authors · 2025-11-27
> **Response to Reviewer 3A4d (1/n)**
>
> Thanks for your insightful review and we would like to address your concerns as follows.
>
> > [Weakness 1, Question 1] Experiments about multiple judges
>
> Thank you for the constructive suggestion. Firstly, We would like to elaborate our choice of reward model: We selected Qwen-Plus as the primary judge model after a comprehensive evaluation of reliability, cost, and latency. As shown in the table below, Qwen-Plus achieves 0.75 human agreement while offering the lowest cost and fastest first-token latency, making it a reliable and cost-friendly option.
>
> | Model             | Agreement | Cost (Input / Output, $/million tokens) | First Token Latency (s) |
> |-------------------|----------|-----------------------------------------|-------------------------|
> | Claude-3.7-Sonnet | 0.82     | 3 / 15                                  | 5.35                    |
> | R1                | 0.76     | 0.57 / 2.29                             | 3.09                    |
> | GPT-4o-2024-11-20 | 0.70     | 2.5 / 10                                | 2.19                    |
> | Qwen-Plus         | 0.75     | 0.4 / 1.2                               | 1.16                    |
>
> Following your constructive suggestion, we conducted additional full training runs with different judges and judge combinations. Results are reported below:
>
> | Model                        | WritingBench | Creative-Writing-Bench | Longbench-Write | Average |
> |-----------------------------------|--------------|------------------------|-----------------|---------|
> | Qwen2.5-7B-WritingBench-SFT	    | 83.71	       | 70.02	                | 92.22	          | 81.98   |
> | Qwen-Plus as Judge                     | 87.23        | 73.19                  | 93.06           | 84.49   |
> | GPT-OSS-120B as Judge                    | 86.75        | 71.60                  | 92.53           | 83.63   |
> | Random Judge (Qwen-Plus, GPT-OSS-120B)   | 87.21        | 71.47                  | 93.99           | 84.22   |
> | Both Judges (Qwen-Plus, GPT-OSS-120B)     | 86.27        | 70.78                  | 93.65           | 83.57   |
>
> * **Replacing Qwen-Plus with a weaker open-source judge GPT-OSS-120B yields reasonable gains but underperforms Qwen-Plus**. Based on our observations, it is likely due to GPT-OSS-120B’s weaker long-form writing evaluation capabilities and over-sensetivity towards minor factual errors in the generated texts, making it hard to comprehensively evaluate overall writing quality.
> * Among combinations, **randomly selecting one of the two judges at each step proves effective and cost-efficient**, nearly matching the performance of using Qwen-Plus alone.
> * In terms of Both Judge setting, it requiring both judges to vote for the generated response to proceed to a higher-quality reference, which may hurt performance. Because any single judge’s bias delays promotion to higher-quality references, slowing curriculum progression especially early in training.
>
> We also evaluated the Output-to-Input generalization performance on LongBench v2. The results are as follows.
>
> | Model                                      | Easy | Hard | Short | Medium | Long  | Overall |
> |--------------------------------------------|------|------|-------|--------|-------|---------|
> | Qwen2.5-7B-Instruct                        | 31.8 | 28.3 | 38.9  | 26.0   | 21.3  | 29.6    |
> | Qwen2.5-7B-WritingBench-SFT                | 27.6 | 27.7 | 35.0  | 25.1   | 20.4  | 27.6    |
> | Qwen-Plus as Judge                         | 35.8 | 29.3 | 42.1  | 25.7   | 26.5  | 31.8    |
> | GPT-OSS-120B as Judge                      | 33.3 | 31.5 | 40.6  | 27.9   | 26.9  | 32.2    |
> | Both Judges (Qwen-Plus, GPT-OSS-120B)      | 32.8 | 32.2 | 42.8  | 26.0   | 27.8  | 32.4    |
> | Random Judge (Qwen-Plus, GPT-OSS-120B)     | 32.8 | 30.9 | 40.0  | 27.4   | 25.9  | 31.6    |
>
> Our findings indicate that the model trained with GPT-OSS-120B demonstrates relatively strong generalization capabilities. Notably, **configurations employing a Random Judge or Both Judges also achieve competitive generalization performance**, suggesting that combining diverse judging strategies holds potential for enhancing model generalization performance.
>
> Finally, we would like to politely point out that **using a single reward model remains common and effective in current LLM training practices**, as evidenced by ChatGPT RLHF[1], DeepSeek reward scaling[2], and Kimi K2 agentic model alignment[3], indicating that a single reward model can provide reliable and effective supervision. Therefore, we believe the single reward model choice is common and therefore not a unique flaw of our proposed framework considering both the effectiveness and costs. But the multi-judge experiments can serve as a good complement. Thanks for the insightful suggestion.
>
> [1] Training language models to follow instructions with human feedback. Neurips 2022
> [2] Inference-time scaling for generalist reward modeling. arXiv.
> [3] Kimi k2: Open agentic intelligence. arXiv.

---

> > ### Author Response · Authors · 2025-11-27
> > **Response to Reviewer 3A4d (2/n)**
> >
> > > [Weakness 2] More dataset details
> >
> > Thanks for your comments and we are more than willing to present more details about our training and evaluation datasets.
> > In terms of topic coverage: We randomly sampled 500 samples in our training data and each of the testsets (include all if total number is smaller than 500) and we categorized them into 8 major domains which are derived from the taxonomy of the three benchmarks. As shown below, the training distribution covers all evaluated domains broadly and the evaluation benchmarks show diverse topic distributions, contributing a more comprehensive evaluation.
> >
> > | Dataset                                | Academic & Scientific Writing | Business & Financial Writing | Legal & Policy Writing | Literary & Creative Writing | Educational & Instructional Writing | Marketing & Communications Writing | Technical & Operational Documentation | Others |
> > |----------------------------------------|-------------------------------|------------------------------|------------------------|-----------------------------|-------------------------------------|------------------------------------|----------------------------------|-----------------------------------|
> > | Training Data         | 17.00                         | 14.50                        | 13.75                  | 33.75                       | 6.75                                | 5.00                                           | 5.75                                  | 2.75 |
> > | WritingBench         | 11.50                         | 14.00                        | 17.50                  | 16.75                       | 19.25                               | 12.25                                   | 4.25                                  | 4.00 |
> > | Longbench-Write       | 17.50                         | 5.83                         | 1.67                   | 33.33                       | 15.83                               | 4.17                                          | 7.50                                  | 14.17 |
> > | Creative-Writing-Bench | 0.00                          | 0.00                         | 0.00                   | 100.00                      | 0.00                                | 0.00                               | 0.00                 | 0.00                                  |
> >
> > To rule out the potential prompt overlap: We conduct a token-level comparsion between the training dataset and the three evaluation benchmarks. The exact match rate is 0 and the n-gram overlap checks (n = 4 to 7) show near-zero overlap (highest 4-gram overlap: 3.13%, dropping rapidly to 0.18% at 7-gram). Therefore, we believe that no identical or near-identical prompts exist between training and any test set.
> >
> > | N-gram Length | Overlap Rate (%) |
> > |---------------|------------------|
> > | 4             | 3.13%            |
> > | 5             | 1.15%            |
> > | 6             | 0.43%            |
> > | 7             | 0.18%            |
> >
> > > [Weakness 3] Variance Calculation
> >
> > Thank you for the constructive suggestions! We have conducted 3 independent experiments for our method, the most competitive baseline in our main experiment and several key ablations. We now report both mean and standard deviation across these runs (represented as mean ± std).
> >
> > | Model                              | WritingBench     | Creative-Writing-Bench | Longbench-Write  | Average         |
> > |------------------------------------|------------------|------------------------|------------------|-----------------|
> > | Qwen2.5-7B-WritingBench-SFT        | 83.71 ± 0.10     | 70.24 ± 1.14           | 92.56 ± 0.80     | **82.17 ± 0.18**    |
> > | Mixed Training (w/o Scheduling)    | 86.51 ± 0.79     | 71.47 ± 0.97           | 92.26 ± 1.24     | **83.41 ± 0.23**    |
> > | Static Scheduling                  | 86.40 ± 0.16     | 72.84 ± 0.98           | 92.26 ± 0.83     | **83.83 ± 0.54**    |
> > | Qwen2.5-7B-Writing-RL (Ours)       | 87.28 ± 0.04     | 73.51 ± 0.74           | 92.64 ± 0.55     | **84.48 ± 0.37**    |
> >
> > As shown in the tables, the updated results confirm that our key contributions are statistically robust:
> > * For curriculum scheduling, our full Writing-RL model with dynamic scheduling reaches 84.48 ± 0.37, substantially higher than Static Scheduling (83.83 ± 0.54) and the no-scheduling ablations (Mixed Training: 83.41 ± 0.23).
> > * For comparison with the competitive baseline, Our model improves with a consistent and relatively large margin.
> >
> > These results, now presented with standard deviations, demonstrate that **the observed improvements are consistent and statistically significant**. We will add more stastical results in our next version and we hope our efforts help to address your concerns.

---

> > > ### Author Response · Authors · 2025-11-27
> > > **Response to Reviewer 3A4d (3/n)**
> > >
> > > > [Weakness 4, Question 2] Cost analysis
> > >
> > > Thanks for your suggestion and we are willing to include more details about the cost of our method to help the community better evaluate the feasibility of reproduction. We would like to politely remind the cost analysis is included in Appendix A.4 and a comprehensive calculation is as follows. We calculate the costs of API calls for close-sourced models and the GPU Hours for the open-source models.
> > >
> > > **Stage 1: Data selection**
> > >
> > > | Model           | Input Price ($/M tokens) | Output Price ($/M tokens) | Cost ($) | GPU Hours (A100) |
> > > |----------------------|--------------------------|---------------------------|----------|-----------|
> > > | Claude-3.7-Sonnet    | 3.00                     | 15.00                     | 375.52   |      --     |
> > > | GPT-4o               | 2.50                     | 10.00                     | 164.33   |     --       |
> > > | Qwen-Plus            | 0.40                     | 1.20                      | 27.74    |     --       |
> > > | Deepseek-R1          | 0.57                     | 2.29                      | 31.12    |     --       |
> > > | Qwen2.5-7B-Instruct-sft  |              --             |              --              |      --     | 34.51     |
> > > | Qwen2.5-7B-Instruct-judge  |              --             |              --              |      --     | 41.25     |
> > > | **Total**             |            --               |            --                | **608.71** | **75.76** |
> > >
> > > We would like to point out that **this is a one-time offline cost that produces a reusable high-quality dataset shareable with the community**. Generating competitive reference responses required calls to four different models (Claude-3.7-Sonnet, GPT-4o, Qwen-Plus, Deepseek-R1), totaling $608.71 in API costs. We noticed that the largest portion came from Claude-3.7-Sonnet and we recommend replacing it with more recent cost-effective alternatives. Additionally, inference with the policy model for references took 34.51 A100 hours, and local scoring of all candidates with a 7B judge model consumed 41.25 A100 hours. Once built, the dataset eliminates these expenses for all subsequent training runs and community reproductions.
> > >
> > > **Stage 2: RL training**
> > >
> > > | Model           | Input Price ($/M tokens) | Output Price ($/M tokens) | Cost ($) | GPU Hours (8*A100) |
> > > |----------------------|--------------------------|---------------------------|----------|-----------|
> > > | Qwen-Plus API           | 0.40                     | 1.20                      | 53.50    |     --       |
> > > | Model Training  |              --             |              --              |      --     | 24     |
> > >
> > > During training process, we used Qwen-Plus as the judge model. A complete training run incurred $53.50 in API costs for reward assignment and required approximately 24 hours on 8×A100 GPUs (including time waiting for judge responses). To effectively reduce our training costs, we have conducted a comprehensive survey to find a reliable and cost-friendly judge model. As shown in the table, Qwen-Plus has already achieved a high agreement with human judges, demonstrating its reward-giving capablities and making it a reliable choice for the training writer models. And Qwen-Plus has a remarkbly lower price than GPT-4o and Claude-3.7-Sonnet and possesses the lowest first token latency, which effectively reduces the API cost burden.
> > >
> > > | Model             | Agreement | Cost (Input / Output, $/million tokens) | First Token Latency (s) |
> > > |-------------------|----------|-----------------------------------------|-------------------------|
> > > | Claude-3.7-Sonnet | 0.82     | 3 / 15                                  | 5.35                    |
> > > | R1                | 0.76     | 0.57 / 2.29                             | 3.09                    |
> > > | GPT-4o-2024-11-20 | 0.70     | 2.5 / 10                                | 2.19                    |
> > > | Qwen-Plus         | 0.75     | 0.4 / 1.2                               | 1.16                    |
> > >
> > > Thus, while the full pipeline does involve repeated judge calls, **the majority of the monetary cost is front-loaded and one-time, and the per-run training cost remains moderate ($53.50 + 24 hours of 8×A100). We believe this level of expenditure is reasonable given the performance gains achieved and the reusability of the constructed resources.**

---

> > > > ### Author Response · Authors · 2025-11-27
> > > > **Response to Reviewer 3A4d (4/n)**
> > > >
> > > > > [Question 3] Ablate curriculum vs RL
> > > >
> > > > Thank you for your constructive questions and we conduct the training experiments as you suggested.
> > > >
> > > > | Model                              | WritingBench | Creative-Writing-Bench | Longbench-Write | Average |
> > > > |------------------------------------|--------------|------------------------|-----------------|---------|
> > > > | Qwen2.5-7B-WritingBench-SFT        | 83.71        | 70.02                  | 92.22           | 81.98   |
> > > > | Qwen2.5-7B-Writing-RL (Ours)       | 87.23        | 73.19                  | 93.06           | 84.49   |
> > > > | Pointwise Reward + No Scheduling   | 86.59        | 70.80                  | 91.25           | 82.88   |
> > > > | Dynamic References SFT             | 84.72        | 67.27                  | 90.83           | 80.94   |
> > > >
> > > > As the results shown, **the components in our framework are essential**: removing scheduling and pairwise reward while keeping PPO yields relatively smaller performance gains, and replacing RL with SFT on the same scheduled references leads to performance drop, especially hurts creative writing. This is because part of the RL data is filtered and selected from the SFT data, therefore continually imitating the responses to the same instructions may cause overfitting and therefore hurt creative writing capabilities.
> > > >
> > > > > [Question 4] Generalization controls
> > > >
> > > > Thank you for the insightful suggestion. To better understand the source of the observed output-to-input generalization, we report the three suggested experiments on LongBench v2:
> > > >
> > > > | Model                                              | Easy | Hard | Short | Medium | Long | Overall |
> > > > |----------------------------------------------------|------|------|-------|--------|------|---------|
> > > > | Qwen2.5-7B-Instruct                                | 31.8 | 28.3 | 38.9  | 26.0   | 21.3 | 29.6    |
> > > > | Qwen2.5-7B-WritingBench-SFT                        | 27.6 | 27.7 | 35.0  | 25.1   | 20.4 | 27.6    |
> > > > | Qwen2.5-7B-Writing-RL (Ours)                       | 35.8 | 29.3 | 42.1  | 25.7   | 26.5 | 31.8    |
> > > > | SFT with RL data                                   | 31.2 | 28.3 | 38.3  | 26.5   | 20.4 | 29.4    |
> > > > | Writing-RL without margin-aware selection          | 32.3 | 30.9 | 40.0  | 26.0   | 27.8 | 31.4    |
> > > > | Writing-RL with static pointwise rewards           | 33.3 | 30.5 | 40.0  | 25.6   | 29.6 | 31.6    |
> > > >
> > > > As the results show, SFT on RL data yields almost no generalization (29.4 < original Qwen2.5-7B-Instruct), confirming that **generalization is not caused by output length overlap or data exposure**. And all the RL-based variants show clear positive generalization, whereas SFT does not. Stronger and more effective RL designs consistently produce larger generalization gains, demonstrating that **the generalization originates from the RL process itself and can be amplified by better RL algorithms and components**.

---

> > > > > ### Author Response · Authors · 2025-11-27
> > > > > **Response to Reviewer 3A4d (5/n)**
> > > > >
> > > > > > [Question 5] Explanation about Figure 2
> > > > >
> > > > > Thanks for your question and we would like to further describe the meaning of the color in Figure 2.
> > > > > **The colors in Figure 2 distinguish different individual samples**. Each colored line tracks the reference-model progression of one specific sample over the course of training. Samples that encounter less challenging writing queries (i.e., queries the model masters relatively quickly) rapidly climb to stronger references early in training (left upper side of the figure). In contrast, samples with more challenging writing queries require more training steps before the LLM consistently outperforms its current reference, resulting in slower, more gradual upward progression (right lower side of the figure). **We aim to use color-coded visualization to illustrate the asynchronous, sample-wise nature of the Dynamic Reference Scheduling mechanism.**
> > > > >
> > > > > > [Question 6] Reference quality
> > > > >
> > > > > Thank you for your insightful question. In the data selection phase, we use an open-source judge model fine-tuned for writing evaluation. This judge assigns each candidate response and generated reference a scalar score on a 1–10 scale, which we use as a proxy for response quality (higher scores represents better quality).
> > > > >
> > > > > To systematically assess the impact of reference quality, we conducted additional experiments using reference sets of varying quality levels.
> > > > >
> > > > > | Reference Quality             | WritingBench Score |
> > > > > |-------------------------------|--------------------|
> > > > > | Baseline (w/o RL)             | 83.71              |
> > > > > | Initial Policy Model          | 86.80              |
> > > > > | Deepseek R1                   | 86.15              |
> > > > > | Highest-quality Reference     | 82.51              |
> > > > > | Dynamic Curriculum (Ours)     | 87.23              |
> > > > >
> > > > > * **When using lower-quality references (e.g., those generated by the initial policy or an earlier version of Deepseek-R1), the model still achieves meaningful gains, but progress slows in later training stages**. The scarcity of high-quality references results in an excessively high density of positive rewards (frequent model wins) as the model evolves, which reduces the learning signal and hinders further improvement after the model achieves a certain level.
> > > > > * Furthermore, **when using only the single highest-scoring reference per writing instruction, performance drops substantially** because the reward signal becomes too sparse: the model rarely beats the reference, making effective learning extremely difficult.
> > > > >
> > > > > We hope our clarifications and efforts help to address your concerns.

---

### Official Review · Reviewer_FD5R · 2025-11-01

**Soundness:** 4
**Presentation:** 3
**Contribution:** 3
**Rating:** 6
**Confidence:** 3

**Summary:**

This paper proposes Writing-RL, an Adaptive Curriculum Reinforcement Learning framework to solve the open-ended long-form writing problem. The framework is built on three key ingredients: (1) A data selection strategy that prioritizes samples with the largest learning potential, which is the score difference between the current policy and the highest-quality reference, (2) A reward function that give rewards via pairwise LLM-as-judge comparison against a high-quality reference, and (3) A dynamic scheduling strategy that assigns reward references of increasing difficulty during training. Writing-RL outperforms a range of strong SFT baselines across all three benchmarks. More interestingly, an “output-to-input” generalization is observed, where models trained with long-output RL also generalize surprisingly well to long-input reasoning tasks.

**Strengths:**

The paper is well motivated and clearly scoped. The writing is well-organized and easy to follow.

The experimental section is comprehensive, solid, and convincing. The authors evaluate against a wide range of competitive baselines across three writing-focused benchmarks, and further include human evaluations to validate the model’s performance. They also report their method’s impact on the general capabilities of LLMs. Finally, the paper contains extensive ablations on curriculum scheduling variants, reward design, and data selection strategies.

The paper also handles experimental details carefully and fairly. For example, it explicitly controls for position bias and justifies the choice of the judge LLM, which increases the credibility of the reported gains.

**Weaknesses:**

A potential weakness is the cost of the overall pipeline. All three stages, data selection, reward assignment, and dynamic scheduling, rely on repeated LLM-as-judge calls, which could make training expensive.

**Questions:**

Algorithm 1 replaces a reference with a stronger one as soon as the model wins once. Did you try requiring k wins before promotion? Given LLM-judge variance, some frontier references may be replaced prematurely (i.e., the model only won once by chance), and the next reference may be too hard, making it impossible for the model to beat it afterward.

---

> ### Author Response · Authors · 2025-11-27
> **Response to Reviewer FD5R (1/n)**
>
> Thanks for your insightful review and we would like to address your concerns as follows.
>
> > [Weakness 1] Cost analysis.
>
> Thanks for your comments and we are willing to include more details about the cost of our method to help the community better evaluate the feasibility of reproduction. We would like to firstly politely remind the cost analysis is included in Appendix A.4 and a comprehensive calculation is as follows. We calculate the costs of API calls for close-sourced models and the GPU Hours for the open-source models.
>
> **Stage 1: Data selection**
>
> | Model           | Input Price ($/M tokens) | Output Price ($/M tokens) | Cost ($) | GPU Hours (A100) |
> |----------------------|--------------------------|---------------------------|----------|-----------|
> | Claude-3.7-Sonnet    | 3.00                     | 15.00                     | 375.52   |      --     |
> | GPT-4o               | 2.50                     | 10.00                     | 164.33   |     --       |
> | Qwen-Plus            | 0.40                     | 1.20                      | 27.74    |     --       |
> | Deepseek-R1          | 0.57                     | 2.29                      | 31.12    |     --       |
> | Qwen2.5-7B-Instruct-sft  |              --             |              --              |      --     | 34.51     |
> | Qwen2.5-7B-Instruct-judge  |              --             |              --              |      --     | 41.25     |
> | **Total**             |            --               |            --                | **608.71** | **75.76** |
>
> We would like to point out that **this is a one-time offline cost that produces a reusable high-quality dataset shareable with the community**. Generating competitive reference responses required calls to four different models (Claude-3.7-Sonnet, GPT-4o, Qwen-Plus, Deepseek-R1), totaling $608.71 in API costs. We noticed that the largest portion came from Claude-3.7-Sonnet and we recommend replacing it with more recent cost-effective alternatives. Additionally, inference with the policy model for references took 34.51 A100 hours, and local scoring of all candidates with a 7B judge model consumed 41.25 A100 hours. Once built, the dataset eliminates these expenses for all subsequent training runs and community reproductions.
>
> **Stage 2: RL training**
>
> | Model           | Input Price ($/M tokens) | Output Price ($/M tokens) | Cost ($) | GPU Hours (8*A100) |
> |----------------------|--------------------------|---------------------------|----------|-----------|
> | Qwen-Plus API         | 0.40                     | 1.20                      | 53.50    |     --       |
> | Model Training  |              --             |              --              |      --     | 24     |
>
> During training process, we used Qwen-Plus as the judge model. A complete training run incurred $53.50 in API costs for reward assignment and required approximately 24 hours on 8×A100 GPUs (including time waiting for judge responses). To effectively reduce our training costs, we have conducted a comprehensive survey to find a reliable and cost-friendly judge model. As shown in the table below, Qwen-Plus has already achieved a high agreement with human judges, demonstrating its reward-giving capablities and making it a reliable choice for the training writer models. And Qwen-Plus has a remarkbly lower price than GPT-4o and Claude-3.7-Sonnet and possesses the lowest first token latency, which effectively reduces the API cost burden.
>
> | Model             | Agreement | Cost (Input / Output, $/million tokens) | First Token Latency (s) |
> |-------------------|----------|-----------------------------------------|-------------------------|
> | Claude-3.7-Sonnet | 0.82     | 3 / 15                                  | 5.35                    |
> | R1                | 0.76     | 0.57 / 2.29                             | 3.09                    |
> | GPT-4o-2024-11-20 | 0.70     | 2.5 / 10                                | 2.19                    |
> | Qwen-Plus         | 0.75     | 0.4 / 1.2                               | 1.16                    |
>
> Thus, while the full pipeline does involve repeated judge calls, **the majority of the monetary cost is front-loaded and one-time, and the per-run training cost remains moderate ($53.50 + 24 hours of 8×A100). We believe this level of expenditure is reasonable given the performance gains achieved and the reusability of the constructed resources.**

---

> > ### Author Response · Authors · 2025-11-27
> > **Response to Reviewer FD5R (2/n)**
> >
> > > [Question 1] Try k wins before promotion
> >
> > Thank you for this highly constructive suggestion and we agree that requiring multiple wins to proceed may mitigate noise from judge variance.
> >
> > Following the suggestion, we conducted an additional experiment with k=2 (i.e., the model must beat the current reference twice before promotion). As shown below, k=2 still yields gains over baselines but underperforms the original k=1 setting.
> >
> > | Model                                   | WritingBench | Creative-Writing-Bench | LongBench-Write | Average |
> > |-----------------------------------------|--------------|------------------|-----------------|---------|
> > | Qwen2.5-7B-Writing-RL (k = 1, original)   | 87.23        | 73.19            | 93.06           | 84.49   |
> > | Qwen2.5-7B-Writing-RL (k = 2)             | 86.98        | 71.49            | 92.36           | 83.61   |
> >
> > Analysis of the training logs reveals that requiring two consecutive wins forces the model to compete with lower-quality references for significantly longer, especially early in training. This substantially delays exposure to high-quality references, reduces overall training efficiency, and ultimately leads to inferior final performance compared to k=1. Regarding the concern that the next reference might be excessively difficult, we measured the average judge score gap between consecutive references across our training dataset and found it to be only 0.3 points on a 10-point scale, which is **moderate and rarely results in an overly large difficulty jump**.
> >
> > Thanks for your insightful suggestions which helps to make this work more solid. We hope our experiments help to address your concerns.

---

### Official Review · Reviewer_CQVx · 2025-11-03

**Soundness:** 2
**Presentation:** 3
**Contribution:** 2
**Rating:** 4
**Confidence:** 4

**Summary:**

The authors propose Writing-RL, an adaptive-curriculum reinforcement-learning framework aimed at improving long-form writing capabilities in large language models. The method is composed of three key components: (i) margin-aware data selection, which prioritizes training examples where the policy model lags significantly behind stronger references; (iii) pairwise comparison reward, which provides more discriminative feedback without verifiable ground truths; (iii) dynamic reference scheduling, which gradually increases task difficulty by upgrading reference quality once the policy surpasses the current baseline.
Empirical results on two backbone models(Qwen2.5‑7B-Instruct and Llama3.1‑8B-Instruct) demonstrate meaningful gains across several long-form writing benchmarks (WritingBench, EQ-Bench, LongBench-Write). The authors additionally observe an unexpected cross-domain effect: models fine-tuned via Writing-RL on long-output generation also show improved performance on long-input reasoning tasks (LongBench-v2), suggesting that RL for long-form generation may also enhance long-context comprehension.

**Strengths:**

1.	The paper tackles long-form writing, a task space where reinforcement learning has struggled due to unverifiable rewards and instability. Extending RL beyond verifiable domains like math or code to open-ended text generation is both challenging and impactful, positioning this work as one of the first systematic studies of RL in long-form writing.
2.	The proposed Writing-RL framework unifies Margin-aware Data Selection, Pairwise Comparison Reward, and Dynamic Reference Scheduling into a cohesive adaptive curriculum. Each addresses a different pain point, data utility estimation, reward discriminability, and difficulty calibration, creating a more stable RL loop for non-verifiable tasks.
3.	The observation that long-output RL improves long-input reasoning on LongBench v2 (Table 3) is insightful. It hints at a potentially deeper connection between generation-length and context-length generalization, offering a new perspective for future long-context training research.

**Weaknesses:**

While the paper presents an interesting direction, several aspects limit its overall contribution and clarity:

1.	Incomplete technical specification.
Many important implementation details are missing, especially in the PPO setup, such as how the value function is parameterized (whether a model like AQuarterMile/WritingBench-Critic-Model-Qwen-7B is used?), whether reward normalization is applied, and what decoding temperature settings are used during training and evaluation. Also mentioned that the paper claims Writing-RL out perform strong SFT baselines but provides no hyperparameter details of SFT experiment(e.g., learning rate, number of epochs, optimizer, or scheduler), making it difficult to verify fairness. Without these, it is difficult to assess reproducibility or confirm whether the reported gains arise from the proposed curriculum design or specific training heuristics.

2.	Limited novelty in core components.
Although the integration of margin-aware data selection and dynamic reference scheduling is well-motivated, the reward design resemble ideas from existing llm-as-a-judge works (e.g., RLAIF by Lee et al. https://openreview.net/forum?id=AAxIs3D2ZZ , JudgeLRM by Chen et al https://arxiv.org/pdf/2504.00050 ). The paper would benefit from a clearer justification or ablation isolating their unique contribution in the writing domain.
3.	Lack of statistical evidence for claimed improvements.
Across multiple ablations (Table 4 for data-selection and Table 6 for curriculum scheduling), the performance gains of the proposed methods are small, often within ~0.5–1.5 point. However, the paper does not report standard deviations. It is therefore unclear whether the improvements of Margin-aware Data Selection over Difficulty-prioritized (86.40 → 87.02) and Dynamic Curriculum over Static/None (84.49 → 83.87/83.15) are statistically significant. The authors should report variance or confidence intervals to substantiate the significance of these differences.
4.	Insufficient discussion of generalization mechanisms.
The “output-to-input generalization” finding is intriguing but currently speculative. The paper lacks controlled experiments to disentangle whether this transfer effect arises from the adaptive curriculum, the LLM-judge preference distribution, or incidental overlap in data distribution.

**Questions:**

1.	Clarification on PPO Implementation and Value Function: The paper states that Writing-RL adopts PPO for training (Section 4.2, Algorithm 1), but the formulation of the value function or advantage estimation is not specified. Could the authors clarify how the critic or value head is parameterized and trained, and whether any reward normalization or bootstrapping strategy is applied given the discrete {0, 0.5, 1} reward signal?
2.	Generalization to Long-Input Reasoning: The “output-to-input generalization” finding is intriguing. Could the authors discuss whether the improvement on long-input reasoning is purely due to the curriculum RL process, or whether the LLM judge’s preference distribution indirectly regularizes long-context coherence? Quantitative ablations on a smaller scale (e.g., replacing the LLM judge to isolate reward-model bias) would strengthen the claim.
3.	Reward Design and Learning Stability: The paper defines a ternary pairwise reward in {0, 0.5, 1}, where 0.5 corresponds to a tie. Could the authors discuss what would happen if the reward were binary only ({0, 1}), without the tie case (1 if Judge(ref, x) = x ≻ ref, and 0 otherwise)? In particular, how would removing the neutral signal affect training stability, variance of policy gradients, or convergence speed in PPO? Some empirical or theoretical justification for including the tie outcome would help clarify its necessity.
4.	Missing hyperparameter details: The paper omits rollout hyperparameters and learning rate scheduler type. Providing these details would improve clarity and reproducibility.
5.	Ablation without value function (GRPO-style baseline): Since the paper attributes Writing-RL’s improvement partly to its adaptive curriculum design rather than the underlying PPO framework, would the authors consider running an ablation that removes the value function entirely, like GRPO-style optimization, while keeping the same LLM-as-a-judge pairwise reward? This experiment could help isolate whether the gains primarily stem from the curriculum mechanism or from the critic-based advantage estimation.

---

> ### Author Response · Authors · 2025-11-27
> **Response to Reviewer CQVx (1/n)**
>
> Thanks for your insightful review and we would like to address your concerns as follows.
>
> > [Weakness 1, Question 1, Question 4] Include details about PPO training setup and SFT parameters.
>
> Thanks for your suggestions and we recognize that full transparency on these aspects is essential for reproducibility. We include some crucial parameter settings in Appendix A.2 and we would like to complement more details as follows.
>
> Regarding the PPO setup, the critic model shares the same backbone as the policy model and is initialized with identical parameters. Since we use the bounded, discrete values (0, 0.5, 1) inherently have relatively low variance with upper and lower bounds, reducing the need for reward normalization. Therefore, we did not apply reward normalization during training. Additionally, as included in Appendix A.2, we use generalized advantage estimation (GAE) as the advantage estimator with discount factor $\gamma = 1.0$ and GAE parameter $\lambda = 1.0$. As for the learning rate, we set a linear warm-up phase to stablize training: a low learning rate (1e-6) with a warm-up ratio of 0.4 for the actor model, while the critic adopts a higher learning rate (1e-5) with a warm-up ratio of 0.05.During rollouts, the sampling temperature was set to $1.0$ throughout training to promote exploration. During evaluation, the temperature is reduced to $0.1$ to ensure stable outputs while reserving a certain level of creativity. Additionally, in Creative-Writing-Bench which focuses on creativity, the temperature is set to 0.7 and the final result is averaged within 3 independent runs, aligned with the original script of the benchmark to provide a fair comparison.
>
> In terms of the SFT parameters, we initially omitted detailed hyperparameters since SFT is not part of the core contribution of this work. Now we would like to complement more details about SFT process: The models were trained using the AdamW optimizer with $\beta_1 = 0.9$, $\beta_2 = 0.999$, and $\epsilon = 1 \times 10^{-8}$. The base learning rate was $7 \times 10^{-6}$, and a cosine decay scheduler with a warmup ratio of $0.1$ (i.e., 10% of total training steps) was applied. Training ran for $5$ epochs with a per-device batch size of $1$ and gradient accumulation over $4$ steps, yielding an effective batch size of $128$ across $32$ GPUs.
>
> > [Weakness 2] Novelty in core components.
>
> Thanks for your comments! We fully acknowledge that prior works, including RLAIF and JudgeLRM as you mentioned, have demonstrated that LLMs can serve as reliable judges and achieve high agreement with human preferences through different prompting strategies, model scales, and further RL-based judge training. These studies primarily focus on improving the accuracy and human agreement of the judge itself. However, our work explores a different direction: **leveraging an LLM judge to enable effective reinforcement learning for open-ended long-form writing, and exploring to expand the recent success of RLVR to the non-verifiable long-form writing domain.** While we adopt pairwise preference scoring from the LLM-as-a-judge paradigm, we incorporate several designs that are critical for successful RL in long-form writing.
> * sample-based multi-dimensional scoring tailored to the open-ended nature of writing tasks;
> * a deliberate exploitation of the judge’s positional bias through our position-disadvantage strategy, which systematically and continually pressures the model to generate higher-quality responses under a given positional disadvantage;
> * an empirical comparison showing that pairwise scoring outperforms pointwise scoring in the writing domain.
>
> Moreover, we would like to point out that our core contribution is the complete adaptive RL framework which consists of margin-aware data selection, pairwise rewarding and dynamic reference scheduling, and the reward design is only one module within our complete framework. The novelty not only lies in the individual design of each module, but also the effective coordination of the three modules to form a performant long-form writing RL recipe.

---

> > ### Author Response · Authors · 2025-11-27
> > **Response to Reviewer CQVx (2/n)**
> >
> > > [Weakness 3] Statistical evidence
> >
> > Thank you for the constructive suggestions! We have conducted 3 independent experiments for the ablations in Table 4 and Table 6 as you mentioned and also the most competitive baseline in our main experiment. We now report both mean and standard deviation across these runs (represented as mean ± std).
> >
> > | Model                              | WritingBench     | Creative-Writing-Bench | Longbench-Write  | Average         |
> > |------------------------------------|------------------|------------------------|------------------|-----------------|
> > | Qwen2.5-7B-WritingBench-SFT        | 83.71 ± 0.10     | 70.24 ± 1.14           | 92.56 ± 0.80     | **82.17 ± 0.18**    |
> > | Mixed Training (w/o Scheduling)    | 86.51 ± 0.79     | 71.47 ± 0.97           | 92.26 ± 1.24     | **83.41 ± 0.23**    |
> > | Static Scheduling                  | 86.40 ± 0.16     | 72.84 ± 0.98           | 92.26 ± 0.83     | **83.83 ± 0.54**    |
> > | Qwen2.5-7B-Writing-RL (Ours)       | 87.28 ± 0.04     | 73.51 ± 0.74           | 92.64 ± 0.55     | **84.48 ± 0.37**    |
> >
> >
> > As shown in the tables, the updated results confirm that our key contributions are statistically robust:
> > * For curriculum scheduling (Table 6), our full Writing-RL model with dynamic scheduling reaches 84.48 ± 0.37, substantially higher than Static Scheduling (83.83 ± 0.54) and the no-scheduling ablations (Mixed Training: 83.41 ± 0.23).
> > * For comparison with the competitive baseline, Our model improves with a consistent and relatively large margin.
> >
> > | Model                 | WritingBench Performance (Mean ± Std) |
> > |-----------------------------------|---------------------|
> > | Difficulty-prioritized Selection   | 86.29 ± 0.28        |
> > | Margin-aware Selection (Ours)      | 86.98 ± 0.23        |
> >
> > * For the data selection ablation experiments(Table 4), Margin-aware Selection achieves 86.98 ± 0.23, significantly outperforming the Difficulty-prioritized baseline of 86.29 ± 0.28.
> >
> > These results, now presented with standard deviations, demonstrate that **the observed improvements are consistent and statistically significant**. We will add more stastical results and include them in our next version in the near future due to computation constraints. We hope our efforts help to address your concerns.

---

> > > ### Author Response · Authors · 2025-11-27
> > > **Response to Reviewer CQVx (3/n)**
> > >
> > > > [Weakness 4, Question 2] More discussion of generalization mechanisms
> > >
> > > Thank you for your recognition and insightful suggestion to further investigate the source of the observed "output-to-input" generalization. To rigorously disentangle the contributing factors, we have conducted a series of controlled experiments that systematically vary the training paradigm (SFT or RL), reward type (pointwise or pairwise), curriculum strategy (none, static or dynamic), and judge model (Qwen-Plus or gpt-oss).
> > >
> > > | Model         | Easy  | Hard  | Short | Medium | Long  | Overall |
> > > |----------------|-------|-------|-------|--------|-------|---------|
> > > | Qwen2.5-7B-Instruct | 31.8 | 28.3 | 38.9 | 26.0  | 21.3 | 29.6    |
> > > | Qwen2.5-7B-WritingBench-SFT (12k) | 27.6 | 27.7 | 35.0 | 25.1  | 20.4 | 27.6    |
> > > | Qwen2.5-7B-Writing-RL (Ours) | 35.8 | 29.3 | 42.1 | 25.7  | 26.5 | 31.8    |
> > > | *Continue SFT with RL data* |       |       |       |        |       |         |
> > > | Qwen2.5-7B-Instruct-Continue-SFT | 31.2 | 28.3 | 38.3 | 26.5  | 20.4 | 29.4    |
> > > | *Curriculum Ablations* |       |       |       |        |       |         |
> > > | Mixed Training (w/o Scheduling)  | 33.3 | 29.6 | 42.8 | 25.1  | 23.1 | 31.0    |
> > > | Static Scheduling     | 32.3 | 30.5 | 38.9 | 25.1  | 30.6 | 31.2    |
> > > | *Judge Model Change*   |
> > > | Qwen2.5-7B-Writing-RL with GPT-OSS-120B as Judge Model | 33.3 | 31.5 | 40.6  | 27.9   | 26.9  | 32.2    |
> > > | *Pointwise Reward* |       |       |       |        |       |         |
> > > | Qwen2.5-7B-Writing-RL with pointwise reward | 33.3 | 30.5 | 40.0 | 25.6  | 29.6 | 31.6    |
> > >
> > > The results show the following clear patterns:
> > > * Training paradigm is the primary driver: continuing supervised fine-tuning (SFT) on exactly the same RL data yields only 29.4 overall (slightly lower than the original Qwen2.5-7B-Instruct), whereas virtually all RL-based variants exhibit positive generalization transfer.
> > > * Judge model has relatively limited impact: replacing our default judge with the open-source alternative GPT-OSS-120B still produces strong generalization (32.2 overall), indicating that the generalization phenomenon is largely **independent of the specific preference distribution of a single judge**.
> > > * Reward design and curriculum design both contribute additional gains: different RL strategies (mixed training without scheduling, static scheduling, pointwise reward design) all display a certain level of generalization, yet **our full method with dynamic curriculum and pairwise preference reward achieves a better result than these ablations.**
> > >
> > > Additionally, to rigorously disentangle the contributing factors and rule out incidental data overlap, we have conducted data analyses about our Writing-RL training dataset and the LongBench-v2 test dataset. We analyze the data distributions in terms of the following aspects.
> > > * Length: The total (input + output) token count in our training data is predominantly below 10k tokens, whereas LongBench v2 contexts range from 8k to over 2M tokens, resulting in minimal length overlap.
> > > * Task Format: Our training data consists exclusively of open-ended long-form writing tasks, while LongBench v2 focuses on long-context understanding via multiple-choice questions.
> > > * Topics: Although a relatively small fraction of writing prompts share broad domains (e.g., finance, law) with some LongBench v2 passages, our RL training framework introduces no additional factual knowledge and seldom conditions on ultra-long relevant contexts, providing no direct knowledge transfer to the LongBench v2 questions.
> > > * Instruction-level Exact Match: We conduct a cross-dataset comparison of each sample's instruction, and we found zero exact match between the datasets.
> > >
> > > These experiments and analysis indicate that **the output-to-input generalization primarily arises from the RL process itself with the choice of training paradigm as the critical factor. Furthermore, it is also shown that fine-grained RL design choices may provide further generalization performance gains**. We hope these experiments help to address your concerns.

---

> > > > ### Author Response · Authors · 2025-11-27
> > > > **Response to Reviewer CQVx (4/n)**
> > > >
> > > > > [Question 3] Reward Design and Learning Stability
> > > >
> > > > Thank you for the thoughtful suggestion. To better understanding the role of tie option, we analyzed all 16,038 preference judgments generated throughout the entire training process and the results are as follows.
> > > >
> > > > | Index   | Reference Wins      | Model Wins   | Tie    | Invalid Format    |
> > > > | ------- | ------ | ------ | ---- | ---- |
> > > > | Count   | 8437   | 7468   | 34   | 99   |
> > > > | Ratio (%)   | 52.61 | 46.56 | 0.21 | 0.62 |
> > > >
> > > > Although we included the tie option to accommodate the open-ended nature of writing evaluation, the judge model is instructed to compare responses across multiple dimensions and then make an overall judgment. In practice, ties are extremely rare (only 0.21% of cases). Model wins (46.56%) and reference wins (52.61%) remain reasonably balanced under dynamic reference scheduling, ensuring efficient learning. **The tie outcome is thus retained merely as a logically reasonable choice but is seldom selected by the judge**. Therefore, removing the tie option and switching to a binary {0, 1} reward would have little noticeable effect on training stability.
> > > >
> > > > > [Question 5] GRPO experiments
> > > >
> > > > Thanks for the comments! To isolate the contribution of the curriculum from the specific advantage estimation in PPO, we conducted an additional ablation using GRPO with the identical pairwise LLM-as-a-judge reward and our full curriculum.
> > > >
> > > > | Model                        | Algorithm | WritingBench | Creative Writing | LongBench-Write | Average |
> > > > |------------------------------|-----------|--------------|------------------|-----------------|---------|
> > > > | Qwen2.5-7B-Writing-RL-PPO    | PPO       | 87.23        | 73.19            | 93.06           | 84.49   |
> > > > | Qwen2.5-7B-Writing-RL-GRPO   | GRPO      | 86.29        | 75.41            | 91.84           | 84.51   |
> > > >
> > > > As shown in the table, GRPO achieves similar average performance (84.51) to PPO (84.49), indicating that **the performance gains are largely agnostic to the specific choice of RL algorithm**.
> > > >
> > > > Thanks for your insightful suggestions which helps to make this work more solid. We hope our experiments help to address your concerns.

---

### Author Response · Authors · 2025-12-03
**Summary of Rebuttal**

We thank the reviewers for the comprehensive reviews and the ACs for the additional efforts under this circumstance. We would like to summary the rebuttal process as follows.

We appreciate that the reviewers acknowledged the contributions of our work: pioneering RL for long-form writing with an adaptive curriculum, strong empirical results with human validation, and insightful generalizations beyond SFT.
- **Reviewers CQVx and GXgG** highlighted the work as one of the first systematic RL studies in the under-explored area of long-form text generation, emphasizing its challenge in unverifiable domains and the intuitive, human-like curriculum design that unifies adaptive components for stability.
- **Reviewers FD5R and TgA2** recognized the well-motivated methods and comprehensive experiments, including extensive ablations on data selection, reward design, and scheduling, alongside careful handling of details like position bias to ensure credible gains on benchmarks.
- **Reviewers 3A4d, CQVx, and TgA2** praised the consistent empirical improvements across writing tasks, competitive with proprietary systems and corroborated by human evaluations, plus valuable insights of Output-to-Input generalization.

## Summary of Key Concerns and Our Responses
1. Key Concern 1: Include details about cost analysis (raised by FD5R, 3A4d, GXgG)

**Our Response:** The majority of the monetary cost is front-loaded and one-time, and the per-run training cost remains moderate ($53.50 + 24 hours of 8×A100). We believe this level of expenditure is reasonable given the performance gains achieved and the reusability of the constructed resources.

2. Key Concern 2: Conduct analysis for statistical evidence (raised by CQVx, 3A4d)

**Our Response:** We conduct more parallel experiments as suggested and the results, presented with standard deviations, demonstrate that the observed improvements are consistent and statistically significant.

3. Key Concern 3: Conduct experiments to better understand the Output-to-Input generalization (raised by CQVx, 3A4d)

**Our Response:** We conduct more controlled experiments and the results indicate that the Output-to-Input generalization primarily arises from the RL process with the choice of training paradigm (SFT or RL) as the critical factor. Furthermore, it is also shown that fine-grained RL design choices may provide further generalization performance gains.

4. Key Concern 4: Address reliability and potential biases of LLM-as-a-judge rewards (raised by 3A4d, TgA2, GXgG)

**Our Response:** We addressed LLM-as-a-judge reliability with a fine-tuned critic model achieving 83% human agreement for data selection and Qwen-Plus at 75% consistency for training. Stability was enhanced by setting temperature to 0, fixing response order to counter positional bias, and specifying evaluation dimensions for deterministic signals. Empirical improvements in writing and generalization validate these strategies' effectiveness despite noise, consistent with standard LLM practices.

## Conclusion
Our work proposes an adaptive RL framework for open-ended long-form writing, exploring to expand the recent success of RLVR to the non-verifiable long-form writing domain. Furthermore, we observe an intriguing generalization from long-output generation to long-input reasoning. We have put in efforts to address the reviewers' concerns and make this work more solid.

---

### Note · Authors · 2026-01-06

**Comment:**

Thanks for the constructive reviews!

**Withdrawal Confirmation:**

I have read and agree with the venue's withdrawal policy on behalf of myself and my co-authors.